# `MiCo`: Multi-image Contrast for Reinforcement Visual Reasoning

**Xi Chen**[1]    **Mingkang Zhu**[3]    **Shaoteng Liu**[3]    **Xiaoyang Wu**[1]    **Xiaogang Xu**[3]
**Yu Liu**[2]    **Xiang Bai**[4]    **Hengshuang Zhao**[1*]

[1]HKU    [2] Tongyi Lab, Alibaba Group    [3]CUHK    [4]HUST

## Abstract

This work explores enabling Chain-of-Thought (CoT) reasoning to link visual cues across multiple images. A straightforward solution is to adapt rule-based reinforcement learning for Vision-Language Models (VLMs). However, such methods typically rely on manually curated question-answer pairs, which can be particularly challenging when dealing with fine-grained visual details and complex logic across images. Inspired by self-supervised visual representation learning, we observe that images contain inherent constraints that can serve as supervision. Based on this insight, we construct image triplets comprising two augmented views of the same image and a third, similar but distinct image. During training, the model is prompted to generate a reasoning process to compare these images (*i.e.*, determine same or different). Then we optimize the model with rule-based reinforcement learning. Due to the high visual similarity and the presence of augmentations, the model must attend to subtle visual changes and perform logical reasoning to succeed. Experiments show that, although trained solely on visual comparison tasks, the learned reasoning ability generalizes effectively to a wide range of questions. Without relying on any human-annotated question-answer pairs, our method achieves significant improvements on multi-image reasoning benchmarks and shows strong performance on general vision tasks.

## 1 Introduction

Making visual analysis with multiple images is crucial in many real-world applications. For example, we understand actions through sequential images or videos, gain 3D awareness by recognizing multiview images, and analyze events by observing differences between states, *etc*. Although Vision Language Models (VLMs) [2, 8, 19, 15, 1] demonstrate promising capabilities in understanding single images, we find them struggle to link visual cues across multiple images.

Multi-image understanding requires not only identifying fine-grained visual cues but also performing logical reasoning to uncover correspondences and differences among images. Recently, reasoning in language models [12, 16, 31, 27] has been significantly improved through the use of Chain-of-Thought (CoT) prompting, especially when combined with rule-based reinforcement learning [28]. Therefore, a straightforward idea to improve multi-image understanding is to extend this reinforcement learning paradigm to the visual domain. However, GRPO [28] requires constructing question-answer pairs with standard answers to compute rewards, which is particularly challenging for tasks involving fine-grained visual details and complex logic across images.

Instead of focusing on constructing QA pairs, we explore how to incentivize VLMs to perform multi-image reasoning with minimal data preparation cost. Modern VLMs already possess strong perceptual and multimodal capabilities. Meanwhile, recent advances in RL-based single image

---

*Corresponding Author

reasoning [34, 23, 5] suggest that reasoning ability can be effectively acquired with limited data. However, most of these methods still rely on task-specific supervision, such as hand-crafted QA pairs. To reduce the reliance on manual annotations, we draw inspiration from self-supervised visual representation learning [7, 14, 13, 3], where images are used as their own source of supervision. Contrastive learning methods [14, 7], for instance, learn discriminative representations by pulling together features from different views of the same image and pushing them away from those of different images. Guided by this principle, we exploit inherent constraints in images as a supervision signal for reward calculation, and present a novel method, MiCo (**M**ultiple **i**mage **Con**trast).

Specifically, we construct training triplets consisting of two augmentations of the same image and a third, different but similar image with its own augmentation. We prompt the VLM to output the thinking process and make comparisons among these images to answer same/different. Multiple trajectories are sampled per example, and reinforcement learning is applied using advantages computed from the correctness of the final answer. A key aspect of our approach is the design of challenging image comparisons. If negative samples are too distinct, the reasoning is trivial. We address this by sampling frames from the same video or using image editing datasets to find similar images, ensuring subtle differences that require careful visual inspection and reasoning. Beyond this contrastive framework, we also introduce *Augmented GRPO*, a training strategy that samples trajectories using weak augmentations and optimizes them under stronger augmentations. This design allows high-quality CoTs to generalize to more difficult images.

Although the model is trained solely on the image comparison task, the learned ability to link visual cues across multiple images generalizes to a wider scope of scenarios. For example, the model can predict plausible future actions by analyzing visual changes across frames, distinguish object identities by comparing fine-grained appearance details, or detect subtle camera movement in scene transformations. Moreover, the contrastive learning process encourages attention to fine-grained details, which also benefits certain single-image understanding tasks like fine-grained layout/attribute understanding. Experimental results show that MiCo achieves strong performance for multi-image understanding [40, 32, 10], and also brings improvements on general vision tasks [11, 6, 39].

## 2 Related Work

**Vision language model reasoning.** Recent studies show that reasoning-capable LLMs [16, 12, 31, 27] can be effectively guided to generate long CoT [36] reasoning processes through reinforcement learning, leading to significant progress on tasks involving complex logic. Building on these advances, a surge of recent works [5, 29, 26, 25] has extended CoT reasoning into the vision-language domain. For example, MM-Eureka [23] expands training data coverage across domains and refines RL training strategies. NoisyRollout [20] introduces image augmentations to enrich the exploration space for policy optimization. LVAA-Thinking [4] provides a detailed analysis of supervised fine-tuning and RL for visual reasoning, along with a curated dataset for related tasks. ThinkLite [34] further improves data efficiency via sample selection with Monte Carlo Tree Search. While these methods rely heavily on curated training data generated by existing models or human annotations, our work explores an alternative: leveraging inherent constraints within visual data to naturally elicit reasoning ability—without explicit question-answer supervision.

**Multi-image understanding.** Understanding multiple images is crucial in real-world scenarios that require comparing object states, tracking actions, or recognizing objects across views. Recent large VLMs [2, 18, 17, 1, 15] have begun to support multi-image inputs natively. LLaVA-Interleave [19] extends LLaVA [19] to process interleaved multimodal inputs. VISC [41] introduces focus-centric data to enhance visual reasoning. Meanwhile, numerous benchmarks [32, 10, 24, 21, 44, 37] have been proposed to evaluate multi-image understanding from various angles. Despite these developments, recent evaluations [40, 46] highlight persistent limitations: VLMs often fail to link fine-grained visual cues across images, such as identifying the same object under different views or detecting subtle state changes for predictive reasoning. Our work addresses this gap by incentivizing the model to compare the fine details across images and make logical analysis.

## 3 Method

### 3.1 Pilot Study for Multi-image Understanding

We begin with a pilot study to assess how well current VLMs understand multiple images. As shown in Fig. 1, we present examples that highlight the capabilities of several state-of-the-art VLMs, Qwen2.5-

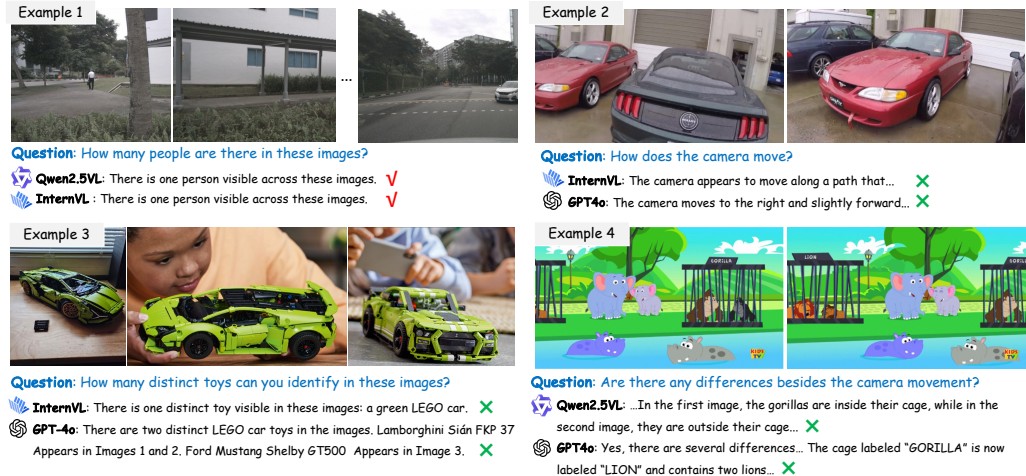

Figure 1: **Challenges for multi-image understanding.** While recent works support multiple images as input, most of them focus on scenarios where each image can be interpreted independently (*e.g.*, Example 1), which remains relatively easy for current state-of-the-art VLMs. However, many real-world tasks (*e.g.*, Example 2-4) require models to compare subtle visual differences, align visual cues across images, and reason about object correspondences—capabilities that current VLMs still struggle with. We gather Example 1 from MuirBench [32], Example 3 from VLM2-bench [40], Example 2,4 from real world samples.

VL [33], InternVL [8], and GPT-4o [15]. While many recent models and benchmarks [24, 32, 44] support multi-image or video inputs, they primarily focus on scenarios like Example 1, where each image can be understood in isolation. In Example 1, models correctly identify a single person across images, reflecting solid basic perception. However, when we examine more complex cases in Fig. 1, we observe that VLMs often suffer from severe hallucinations. In Example 2, both models fail to infer the correct camera movement, showing weaknesses in spatial reasoning. In Example 3, VLMs cannot distinguish between different car toys, indicating difficulty with cross-image comparison. Example 4 further reveals failure in tracking semantic changes across images, with hallucinated object positions and misidentified labels. These cases highlight that current VLMs still lack robust visual comparison abilities essential for multi-image understanding. These examples typically require the model to explicitly link visual cues across images, analyze fine-grained differences, and reason about inter-image correspondences.

As current VLMs already possess strong abilities in single-image perception (*e.g.*, reading fine-grained text) and demonstrate solid commonsense knowledge, as evidenced by their performance on standard vision benchmarks. We hypothesize that their primary limitation in multi-image understanding lies in their inability to compare and connect visual information across images. To address this gap, we focus on enhancing the meta-cognitive ability of **visual comparison**, the core skill needed for effective multi-image reasoning.

### 3.2 Multi-image Contrast

Rather than collecting data for each specific multi-image task, we aim to improve VLMs' general capacity to analyze and reason over multiple images by targeting the core meta skill: **visual comparison**. Inspired by the principles of self-supervised learning, we design a lightweight and scalable framework that encourages the model to distinguish similar yet distinct images. By simulating contrastive visual situations and prompting the model to generate structured reasoning trajectories, we aim to enhance its ability to perceive fine-grained differences, establish correspondences, and perform step-by-step comparisons across images.

Here, we elaborate on the pipeline of `MiCo`. The overall framework consists of the following main steps. First, we identify and construct contrastive image samples that are visually similar yet different. Then, we apply data augmentation to build informative training triplets. Finally, we leverage Augmented GRPO to evaluate a set of reasoning trajectories and optimize the VLM accordingly.

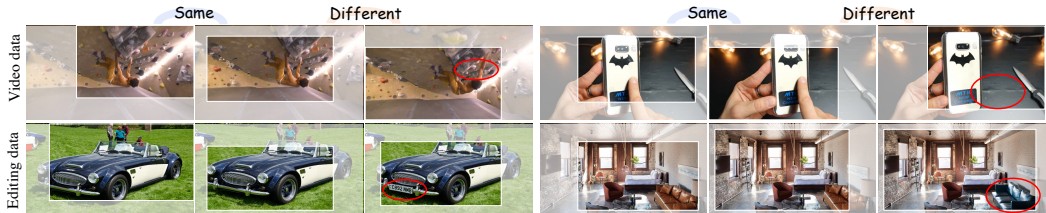

Figure 2: **Demonstrations for contrastive samples.** The first row shows two triplets from the video, and the second row demonstrates samples from image editing datasets. These samples are visually similar but contain subtle differences (marked with red circles), on which we apply random cropping and resizing. In each triplet, the first two images are the same, and the third image is different.

**Image selection.** We begin by selecting image pairs that are visually similar but exhibit subtle differences, which serve as contrastive supervision signals. We denote such a pair as $(I_a, I_b)$, where $I_a$ and $I_b$ are distinct images sharing high structural similarity (*e.g.*, similar layout or background), but with small detail variations.

We leverage two types of data sources that naturally fulfill this requirement: video frames and image editing datasets. For video data, we randomly sample $(I_a, I_b)$ from the same video with a temporal gap of 2 seconds, and compute their Structural Similarity (SSIM) to filter out near-identical pairs. For image editing data, each $(I_a, I_b)$ pair consists of a "before" and "after" edited image. We compute the pixel-wise Mean Squared Error (MSE) to remove significantly different pairs. These constraints ensure that the collected pairs exhibit subtle but meaningful changes.

**Image augmentation.** While the selected image pairs already exhibit subtle variations, directly learning to distinguish them may still lead to shortcut learning. To increase task complexity and encourage detailed reasoning, we apply data augmentation to create diverse image views.

As the visualization examples provided in Fig. 2, given a source image $I$, we generate two augmented versions via random cropping and resizing (they do not change the content of images). For each image pair $(I_a, I_b)$, we thus construct a contrastive triplet:

$$\mathcal{T} = \{\mathcal{T}_1(I_a), \mathcal{T}_2(I_a), \mathcal{T}_3(I_b)\},$$

### 3.3 Augmented GRPO

**Question-answer formulation.** After getting the triplets that contain similar images and their augmentations. We construct QA pairs for reinforcement learning. Given an image triplet, we add reasnoning prompt and user questions as follows:

---

**Reasoning Template of MiCo**

**Reasoning Prompt:** First output the thinking process in <think> </think> and give the final answer in <answer> </answer> tags.

**User Question:** Regardless of the augmentation, are image1 and image2 the same? How about image2 and image3, image1 and image3? Only return T(True) or F(False) in <answer> </answer>, for example <think> </think> <answer>TFT</answer>.

---

To increase the diversity and balance the difficulties of questions, besides the image triplet, we also construct image pairs and design the corresponding prompts for comparing two images. In addition, we use GPT-4o [15] to expand the user question of the same meaning but with various expressions.

**Rollout augmentation.** For each question $q$ with augmented images $\mathcal{T}$, the original GRPO samples a group of outputs $\{o_1, o_2, \cdots, o_G\}$ from the old policy $\pi_{\theta_{old}}$ and then optimizes the policy model $\pi_\theta$. To better leverage difficult samples that arise from strong augmentations, we sample trajectories using weakly augmented inputs $\mathcal{T}^w$, which are easier to produce valid reasoning chains. These sampled trajectories are then used to optimize the policy on stronger augmented contexts $\mathcal{T}^s$, effectively transferring reliable behavior to harder instances.

---

**Algorithm 1** `MiCo`: Reinforcement Multi-image Reasoning

---
1: **Input:** Policy $\pi_\theta$, old policy $\pi_{\theta_{\text{old}}}$, image triplet dataset $\mathcal{D} = \{(I_1, I_2, I_3)\}$, training steps $T_{\max}$, group size $G$, clip parameter $\epsilon$, weak augment operators $\mathcal{T}^{\text{w}}$, strong augment operators $\mathcal{T}^{\text{s}}$
2: **for** $t = 1$ to $T_{\max}$ **do**
3:      Sample triplet $(I_1, I_2, I_3) \sim \mathcal{D}$
4:      Apply weak augmentation: $(I_1^w, I_2^w, I_3^w) = \mathcal{T}^{\text{w}}(I_1, I_2, I_3)$
5:      Apply strong augmentation: $(I_1^s, I_2^s, I_3^s) = \mathcal{T}^{\text{s}}(I_1, I_2, I_3)$
6:      Construct prompts $\mathbf{q}^w$ and $\mathbf{q}^s$ from the weak and strong augmented triplets, respectively
7:      Sample $G$ CoT responses $\{\mathbf{o}_i\}_{i=1}^G$ from $\pi_{\theta_{\text{old}}}(\cdot \mid \mathbf{q}^w)$          ▷ Rollouts from weak prompt
8:      Evaluate reward $R_i = R(I^w, \mathbf{q}^w, \mathbf{o}_i)$ for each $i = 1, \ldots, G$
9:      Compute group baseline $\bar{R} = \frac{1}{G} \sum_{i=1}^G R_i$, and advantages $\hat{A}_i = \frac{R_i - \bar{R}}{\sigma(R)}$
10:     Optimize $\pi_\theta$ on the strong prompt $\mathbf{q}^s$ using the group rollouts:
11:     $L(\theta) = \frac{1}{G} \sum_{i=1}^G \min\left(r_i \hat{A}_i, \text{clip}(r_i, 1-\epsilon, 1+\epsilon)\hat{A}_i\right)$, where $r_i = \frac{\pi_\theta(\mathbf{o}_i|\mathbf{q}^s)}{\pi_{\theta_{\text{old}}}(\mathbf{o}_i|\mathbf{q}^s)}$
12:     $\theta \leftarrow \theta - \nabla_\theta L(\theta)$
13:     $\theta_{\text{old}} \leftarrow \theta$
14: **end for**

---

**Training objective.** The training objective of Augmented GRPO could be formulated as follows. This objective encourages the policy to assign higher likelihoods to responses with higher relative rewards within each group.

$$\mathcal{J}(\theta) = \mathbb{E}[q \sim P(Q), \{o_i\}_{i=1}^G \sim \pi_{\theta_{old}}(O|q)]$$
$$= \frac{1}{G} \sum_{i=1}^G \left( \min\left( \frac{\pi_\theta(o_i|q)}{\pi_{\theta_{old}}(o_i|q)} A_i, \ \text{clip}\left( \frac{\pi_\theta(o_i|q)}{\pi_{\theta_{old}}(o_i|q)} \right) A_i \right) - \beta \, \mathbb{D}_{KL}\left( \pi_\theta || \pi_{ref} \right) \right), \quad (1)$$

$$\mathbb{D}_{KL}\left( \pi_\theta || \pi_{ref} \right) = \frac{\pi_{ref}(o_i \mid q)}{\pi_\theta(o_i \mid q)} - \log \frac{\pi_{ref}(o_i \mid q)}{\pi_\theta(o_i \mid q)} - 1, \quad (2)$$

where $\epsilon$ and $\beta$ are hyperparameters. Following GRPO [28], $A_i$ is the normalized advantage computed based on rewards $\{r_1, r_2, \ldots, r_G\}$.

$$A_i = \frac{r_i - \text{mean}(\{r_1, r_2, \cdots, r_G\})}{std(\{r_1, r_2, \cdots, r_G\})}. \quad (3)$$

Following DeepSeek-R1 [12], we leverage the binary format reward and the accuracy reward, which considers the matching of "<think> </think> <answer> </answer>" tags, and the correctness of the final answer. For the triplet comparisons, we get 1 for the accuracy reward only if we make correct comparisons for all three pairs.

**Overall algorithm.** Our `MiCo` could be summarized in Algorithm 1. We first construct an image triplet consisting of two augmented views of the same image and a third, visually similar but distinct image (with augmentations). The model is prompted to perform multi-image comparison and generate reasoning trajectories. During training, chain-of-thought responses are sampled from the weakly augmented views, and the policy is optimized on the strongly augmented ones using rule-based reinforcement learning. This process enables the model to learn fine-grained visual reasoning in a self-supervised manner.

## 4 Experiments

### 4.1 Implementation Details

**Hyper-parameters.** For the baseline model, we follow previous works [23, 20, 34, 4] and select Qwen2.5-VL-7B [2]. For the training data, we use OmniEdit [35] for image editing pairs and extract video frames from Vidgen-1M [30]. The part of reinforcement learning follows GRPO [28], we set a

Table 1: **Performance on VLM2-Bench** [40], which evaluates the ability to compare and link fine-grained visual cues across multiple images. Without relying on any human- or model-annotated data, MiCo achieves significant improvements and sets a new state-of-the-art. Reasoning-based models (marked with ●) are evaluated using their corresponding prompting strategies.

| Baselines or Models | General | | Object | | | Person | | | | Overall* | |
|---|---|---|---|---|---|---|---|---|---|---|---|
| | Mat | Trk | Cpr | Cnt | Grp | Cpr | Cnt | Grp | VID | Avg | $\Delta_{human}$ |
| Chance-Level | 25.00 | 25.00 | 50.00 | 34.88 | 25.00 | 50.00 | 34.87 | 25.00 | - | 32.73 | -61.44 |
| Human-Level | 95.06 | 98.11 | 96.02 | 94.23 | 91.29 | 97.08 | 92.87 | 91.17 | 100.00 | 95.16 | 0.00 |
| ○ LLaVA-OneVision[17] | 16.60 | 13.70 | 47.22 | 56.17 | 27.50 | 62.00 | 46.67 | 37.00 | 47.25 | 39.35 | -55.81 |
| ○ LLaVA-Video-7B [43] | 18.53 | 12.79 | 54.72 | 62.47 | 28.50 | 62.00 | 66.91 | 25.00 | 59.00 | 45.65 | -49.51 |
| ○ LongVQA-7B [42] | 14.29 | 12.98 | 46.53 | 49.47 | 29.00 | 58.00 | 41.56 | 25.00 | 45.00 | 37.10 | -58.06 |
| ○ mPLUG-Owl2-7B [38] | 17.37 | 18.26 | 49.17 | 62.97 | 31.00 | 63.00 | 58.06 | 29.00 | 43.00 | 40.87 | -54.31 |
| ○ Qwen2-VL-7B [2] | 18.07 | 19.18 | 68.08 | 61.84 | 37.50 | 72.00 | 67.92 | 47.00 | 55.25 | 49.76 | -45.40 |
| ○ InternVL2.5-8B [8] | 41.24 | 26.53 | 72.22 | 67.65 | 40.00 | **85.00** | 66.67 | 52.25 | 50.25 | 55.41 | -39.75 |
| ○ InternVL2.5-26B [8] | 30.50 | 30.59 | 43.33 | 51.48 | 52.50 | 59.50 | 59.67 | 61.25 | 45.25 | 45.59 | -49.57 |
| ○ Qwen2.5-VL-7B [2] | 35.91 | 43.38 | 71.39 | 41.72 | 47.50 | 80.00 | 59.76 | 69.00 | 45.00 | 54.82 | -40.34 |
| ○ GPT-4o [15] | 37.45 | 39.27 | 74.17 | **80.62** | 57.50 | 50.00 | **90.50** | 47.00 | **66.75** | 60.36 | -34.80 |
| ● MM-Eureka-7B [23] | 55.60 | 47.03 | 74.10 | 52.50 | 54.00 | 77.50 | 60.00 | 51.00 | 43.50 | 57.24 | -37.91 |
| ● NoisyRollout-7B [20] | 40.93 | 43.83 | 63.33 | 50.83 | 34.50 | 70.50 | 63.33 | 47.00 | 36.50 | 50.08 | -45.08 |
| ● ThinkLite-VL-7B [34] | 40.45 | 46.58 | 75.56 | 62.50 | 49.50 | 77.50 | 62.50 | 51.00 | 36.50 | 55.79 | -39.37 |
| ● VLAA-Thinker-7B [4] | 47.49 | 63.03 | 72.20 | 61.40 | 55.00 | 71.00 | 57.50 | 51.00 | 47.75 | 58.49 | -36.67 |
| ○ Qwen2.5-VL-7B-CoT[2] | 43.24 | 42.92 | 66.39 | 50.56 | 36.00 | 62.50 | 55.83 | 39.00 | 36.75 | 48.91 | -46.24 |
| ● MiCo-7B-CoT | **57.14** | **67.12** | **81.94** | 56.67 | **58.00** | 65.00 | 57.50 | 62.00 | 44.25 | **61.06** | **-34.09** |
| Δ Improvement | +13.90 | +24.20 | +15.55 | +6.11 | +22.00 | +2.50 | +1.67 | +23.00 | +7.50 | +12.93 | +12.93 |

format reward and accuracy reward with the weight of 1:1, respectively. Besides, we also apply a KL regularization with a weight of 0.01. During training, we follow previous works [23] to skip the rollout group with all correct/false answers. During training, we use a learning rate of 1e-6 and set the batch size of 16. For each training sample, we generate a group of 8 rollouts. We train the model for 600 iterations on $8 \times$A100 GPUs.

**Evaluation protocols.** For evaluation, we follow the default hyper-parameters of Qwen2.5-VL [33] and utilize the VLMEvalKit [9]. For reasoning baselines, we adopt their official prompting formats. Minor inconsistencies in results may occur due to differences in the implementation details of evaluation frameworks or answer parsing logic.

## 4.2 Result Analysis for Multi-image Comparison

**Evaluation metrics.** We first report the model performance on VLM2-Bench [40]. This benchmark mostly aligns with our intention of linking fine-grained visual cues across images. Specifically, VLM2-Bench includes three tracks: General Cue (GC), Object-centric Cue (OC), and Person-centric Cue (PC). Each track consists of subtasks with specific metrics: Mat (Matching) and Trk (Tracking) use paired T/F accuracy; Cpr (Comparison) evaluates consistency by requiring the model to correctly answer both a positive and its corresponding negative statement; Cnt (Counting) uses normalized error to measure numerical prediction accuracy; Grp (Grouping) is a multiple-choice task assessing clustering ability; and VID (Video Identity Describing) is scored based on GPT-4o evaluation of open-ended descriptions.

**Result analysis.** As shown in Tab. 1, we present the comparison results on VLM2-Bench. We observe that all existing open- and closed-source models lag behind human performance by a large margin. Among them, GPT-4o [15] demonstrated clear advantages over other models. Thanks to the strong generalization ability of reinforcement learning, recent reasoning VLMs [23, 20, 34, 4] have shown consistent improvements when built upon Qwen2.5-VL-7B [2]. We report the performance of MiCo in the final block. Trained with contrastive triplets, MiCo effectively learns the core ability to compare images, achieving substantial gains across multiple tasks and obtaining the best average performance overall. Notably, our 7B model even outperforms GPT-4o. However, we find that CoT reasoning does not benefit all sub-tasks equally. Specifically, for tasks involving human faces (person track), CoT-based models offer limited or even negative gains compared to no-CoT counterparts. We hypothesize that human identity representations, such as facial nuances, are difficult to verbalize, thus limiting the benefit of language-based reasoning. In contrast, object-level identity differences (*e.g.*, logos,

Table 2: **Ablation studies** on key configurations. We conduct experiments on VLM2-Bench [40] and report the average accuracy across the general, object, and person tracks. For each ablation, all other settings are kept consistent with our final model to ensure fair comparisons.

<table>
<tr><td colspan="4" align="center">(a) Learning Paradigm</td></tr>
<tr><td></td><td>General</td><td>Object</td><td>Person</td></tr>
<tr><td>Qwen2.5-VL [2]</td><td>39.64</td><td>53.53</td><td>63.44</td></tr>
<tr><td>SFT</td><td>42.90</td><td>51.15</td><td>55.98</td></tr>
<tr><td>No-CoT RL</td><td>45.36</td><td>50.01</td><td>55.23</td></tr>
<tr><td>CoT RL</td><td>62.13</td><td>65.53</td><td>57.18</td></tr>
</table>

<table>
<tr><td colspan="4" align="center">(b) Data Source</td></tr>
<tr><td></td><td>General</td><td>Object</td><td>Person</td></tr>
<tr><td>Edit Data[1] [35]</td><td>61.23</td><td>65.33</td><td>56.35</td></tr>
<tr><td>Edit Data[2] [45]</td><td>60.88</td><td>64.33</td><td>55.27</td></tr>
<tr><td>Video Data</td><td>60.29</td><td>64.50</td><td>55.68</td></tr>
<tr><td>Edit[1] + Video</td><td>62.13</td><td>65.53</td><td>57.18</td></tr>
</table>

<table>
<tr><td colspan="4" align="center">(c) Rollout Augmentation</td></tr>
<tr><td></td><td>General</td><td>Object</td><td>Person</td></tr>
<tr><td>Qwen2.5-VL [2]</td><td>39.64</td><td>53.53</td><td>63.44</td></tr>
<tr><td>(Strong, Strong)</td><td>59.41</td><td>64.00</td><td>56.98</td></tr>
<tr><td>(Weak, Weak)</td><td>55.58</td><td>62.03</td><td>54.81</td></tr>
<tr><td>(Weak, Strong)</td><td>62.13</td><td>65.53</td><td>57.18</td></tr>
</table>

<table>
<tr><td colspan="4" align="center">(d) Sample Formulation</td></tr>
<tr><td></td><td>General</td><td>Object</td><td>Person</td></tr>
<tr><td>Qwen2.5-VL [2]</td><td>39.64</td><td>53.53</td><td>63.44</td></tr>
<tr><td>Image Pairs</td><td>56.41</td><td>66.98</td><td>55.68</td></tr>
<tr><td>Image Triplets</td><td>60.64</td><td>65.33</td><td>55.81</td></tr>
<tr><td>Pairs + Triplets</td><td>62.13</td><td>65.53</td><td>57.18</td></tr>
</table>

<table>
<tr><td colspan="4" align="center">(e) Prompt Diversity</td></tr>
<tr><td></td><td>General</td><td>Object</td><td>Person</td></tr>
<tr><td>Qwen2.5-VL [2]</td><td>39.64</td><td>53.53</td><td>63.44</td></tr>
<tr><td>Single Prompt</td><td>55.53</td><td>64.16</td><td>51.50</td></tr>
<tr><td>20 Variations</td><td>62.13</td><td>65.53</td><td>57.18</td></tr>
<tr><td>50 Variations</td><td>63.13</td><td>65.29</td><td>54.93</td></tr>
</table>

<table>
<tr><td colspan="4" align="center">(f) Image Augmentations</td></tr>
<tr><td></td><td>General</td><td>Object</td><td>Person</td></tr>
<tr><td>Base (Crop, Resize)</td><td>62.13</td><td>65.53</td><td>57.18</td></tr>
<tr><td>Base + Flip</td><td>61.13</td><td>63.98</td><td>56.77</td></tr>
<tr><td>Base + Rotat.</td><td>62.58</td><td>65.03</td><td>55.86</td></tr>
<tr><td>Base + Color.</td><td>60.15</td><td>64.26</td><td>54.86</td></tr>
</table>

textures, shapes) are more readily describable, allowing CoT reasoning to help reduce hallucinations and improve distinction.

## 4.3 Ablation Studies

We conduct a series of ablation studies to validate the effectiveness of our core designs. As an initial exploration of visual reasoning, we also analyze the impact of some basic configurations.

**Training strategies.** In Tab. 2 (a), we evaluate different training paradigms. We first apply supervised fine-tuning (SFT) on our contrastive dataset, allowing the model to directly predict the final answer. We observe that this leads to minor gains on the general track, which more closely aligns with the training task. However, the ability acquired through SFT does not generalize well to more diverse reasoning tasks. We also test a no-CoT reinforcement learning baseline, where the model is trained to output answers directly using GRPO [28]. Due to the absence of intermediate reasoning steps, the resulting trajectories are short and behave similarly to SFT, yielding limited improvements.

**Data source.** In Tab. 2 (b), we compare different training data sources. Both the image editing data (OmniEdit [35]) and video-derived frames (VidGen [30]) individually support effective learning. Combining these two heterogeneous sources further enhances performance. We also validate that our framework is not tied to specific editing styles, as models trained on either OmniEdit or UltraEdit [45] generalize well, demonstrating robustness to the editing domain variation.

**Rollout augmentation.** In this work, we leverage weak augmentations for rollout sampling, and use these high-quality answers to optimize harder questions with stronger augmentations. In Tab. 2 (c), we report different combinations of augmentation in "(sampling, optimization)" process. We show that, strong augmentations are vital for contrastive learning compared with weak augmentations, and our rollout augmentation strategy gets the best performance.

**Sample formulation.** As discussed in Sec. 3.3, we construct prompts based on either image pairs or image triplets. While we initially suspected that binary image-pair comparisons (with 50% guess probability) might result in low-quality CoTs, our experiments reveal that they still contribute positively to performance. In practice, we find that combining both formats—pair-based and triplet-based leads to the best results.

**Other configurations.** In Tab. 2 (e), (f), we explore the effects of prompt and augmentation diversity. We observe that increasing the variation of image prompts helps prevent overfitting, with performance

Table 3: **Quantitative results on general vision benchmarks.** We report performance for wide scenarios. Multi-image benchmarks are marked in bold. `MiCo` brings steady improvements compared with our baseline, and gets competitive results against other visual reasoning models.

| | **MuirBench** [32] | **BLINK** [10] | Hallusion [11] | MMStar [6] | MMMU [39] | MathVistas [22] |
|---|---|---|---|---|---|---|
| MM-Eureka-7B [23] | 60.57 | 54.39 | 68.45 | 65.73 | 54.11 | 72.00 |
| NoisyRollout-7B [20] | 59.61 | 56.07 | 66.66 | 65.66 | 54.55 | 71.60 |
| VLAA-Thinker-7B [4] | 61.00 | 54.81 | 69.08 | 63.60 | 54.44 | 70.80 |
| ThinkLite-VL-7B [34] | 57.62 | 55.81 | 72.97 | 66.80 | 53.55 | 71.89 |
| Qwen2.5VL-7B [2] | 58.43 | 55.54 | 69.50 | 64.06 | 54.11 | 67.10 |
| MiCo-7B | 60.53 | 57.23 | 69.61 | 65.60 | 54.77 | 67.90 |
| Δ Improvement | +2.10 | +1.69 | +0.11 | +1.54 | +0.66 | +0.80 |

Table 4: **Task analysis for visual reasoning.** We list representative sub-tasks from MuirBench [32] and BLINK [10] to analyze the generalization ability and limitations for `MiCo`.

| | Visual retrieval | Semantic Corr. | Spatial Rela. | Scene Under. | Forensic Det. | Relative Depth |
|---|---|---|---|---|---|---|
| Qwen2.5VL-7B [4] | 63.69 | 33.09 | 88.81 | 61.82 | 48.48 | 81.45 |
| MM-Eureka-7B [23] | 57.19 + | 33.09 + | 82.51 - | 67.74 ++ | 50.00 + | 75.80 - |
| VLAA-Thinker-7B [4] | 68.83 + | 34.53 + | 86.71 - | 69.89 ++ | 47.72 - | 76.61 - |
| MiCo-7B | 71.23 ++ | 41.72 ++ | 90.20 + | 63.97 + | 47.72 - | 78.22 - |

saturating at around 50 distinct prompt templates. For image augmentations, we experimented with various techniques and ultimately selected random cropping and resizing as the default setting based on empirical performance.

## 4.4 Analysis on General Vision Tasks

In this section, we evaluate the generalization ability and capacity boundaries of `MiCo` on a broader range of vision tasks. We first report quantitative results on additional benchmarks and analyze performance across more diverse task types.

**Results on additional benchmarks.** As shown in Tab. 4, we evaluate `MiCo` on MuirBench [32] and BLINK [10], both of which are representative multi-image understanding benchmarks. To further assess generalization, we also include several single-image benchmarks, including MMStar [6], MMMU [39], HallusionBench [11], and MathVista [22].

Compared to methods trained with manually curated supervision, our contrastive learning framework exhibits strong performance on multi-image understanding tasks, where relational reasoning across images is crucial. While `MiCo` also improves over standard baselines on single-image tasks, its performance remains behind models trained with task-specific guidance, particularly in complex scenarios like visual mathematics, where symbolic reasoning and structured representation are essential but not explicitly modeled in our current training paradigm.

**Task-wise analysis.** We further analyze the performance of `MiCo` on specific sub-tasks from MuirBench [32] and BLINK [10] to better understand its strengths and limitations. Our contrastive learning framework demonstrates clear advantages on correspondence-style tasks, such as *Visual Retrieval* and *Semantic Correspondence*, where `MiCo` outperforms other reasoning models. These results highlight the model's strength in aligning multimodal signals through relational comparisons. In addition, *Spatial Relation* tasks—which evaluate the model's understanding of image layout and object positioning—also benefit from contrastive training. By encouraging attention to relative positions among visual entities, `MiCo` achieves the highest accuracy in this category.

On the other hand, `MiCo` lags behind models trained with manually curated reasoning datasets on tasks such as *Scene Understanding* and *Forensic Detection*, which typically rely on single-image question answering. These tasks often demand domain-specific priors or curated logic patterns that are less emphasized in our data construction process. We also observe that *Relative Depth*, a representative spatial reasoning task, remains challenging for all models and exhibits a noticeable performance drop. This suggests that depth-aware understanding is not sufficiently captured by current training signals. Incorporating explicit spatial or geometric cues remains a promising direction for future work toward building spatially grounded vision-language models.

Question: How many different cups could you find?

**MiCo**: To determine how many different cups can be found, let's analyze each image:

*Image 1*: The person is holding a cup with a simple design of coffee beans and steam. This appears to be one type of cup.

*Image 2*: The person is holding a cup with a more intricate design featuring swirls and coffee beans. This looks like a different style of cup compared to the first one.

*Image 3*: The cup here has a similar design to the second image, with swirls and coffee beans…This suggests that this cup might be the same as the one in Image 2.

Image 1 shows one type of cup. Image 2 shows another distinct type of cup. Image 3 shows the same cup as in Image 2.

**Therefore, there are two distinct types of cups visible across all three images.**

Question: How to transform the first image to the second?

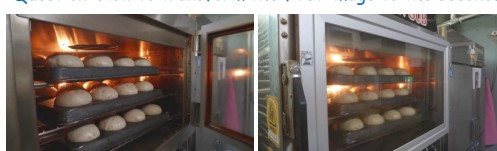

**MiCo**: Let's analyze the differences between the two images:.

*Oven Door*: In the first image, the oven door is open, revealing the interior where the bread is baking. In the second image, the oven door is closed, and we can see the bread through the glass window.

*Lighting*: The lighting inside the oven appears brighter in the first image compared to the second image, which might be due to the open door allowing more light to enter.

*Background Elements*: The second image has additional elements in the background, such as a sign that reads "SUB" and a pink cloth hanging on the wall, which are not present in the first image.

*Focus:* The focus seems to be on the bread in both images, but the second image provides a clearer view of the bread through the closed door.

**Hence, close the door of the oven.**

Q: How does the camera move when shooting the video?

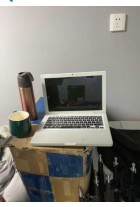
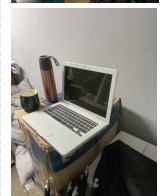

**MiCo**: The differences between the two images suggest that there has been a change in the position of the laptop relative to the rest of the scene, particularly concerning its angle and the position of the items on the table. In the first image, the laptop appears to be more angled towards the viewer, while in the second image, it seems to be more upright and centered. Given these observations, it's likely that the camera moved to capture this change in perspective. Since the first image is from the beginning of the video and the second image is from the end, and considering the changes described.

**Hence, the camera must have moved to the right to achieve this new framing.**

Question: Which point is corresponding to the reference point?

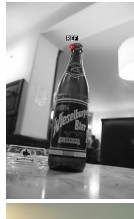

**MiCo**: To determine the corresponding point for the reference point (REF) in the first image, we need to identify a similar point in the second image that could be analogous.

In the first image: The reference point (REF) is located near the top center of the bottle, close to the cap area.

In the second image:
- *Point A* is at the very top of the bottle's cap.
- *Point B* is slightly below the cap, near the neck of the bottle.
- *Point C* is around the middle of the bottle, near the label.
- *Point D* is at the bottom of the bottle.
The point that is closest to the top center of the bottle, similar to where the reference point is located in the first image, is *Point A*.

**Therefore, the answer is *Point A*.**

Figure 3: **Demonstrations for visual reasoning.** Given a question, MiCo first examines the details of each image to identify answer-related visual cues, and then performs cross-image comparisons to derive the final answer. The reasoning processes are marked in gray, with key contents underlined.

## 4.5 Qualitative Analysis

We provide qualitative demonstrations in Fig. 3 to illustrate the multi-image reasoning process of MiCo. For each question, the model first generates a detailed reasoning chain that carefully compares visual cues across the input images. This includes analyzing fine-grained differences and similarities that are relevant to the question. Based on this reasoning, the model then produces a final answer that successfuly addresses the query objective.

## 4.6 Unsuccessful Attempts

Throughout our exploration, we experimented with several alternative approaches that ultimately did not lead to improved performance. For completeness and to facilitate future research, we summarize these unsuccessful attempts and provide insights into why they may have failed.

**Confidence reweighting.** Since our task is formulated as answering T/F questions, even when evaluating three comparisons simultaneously, there remains a non-trivial chance (12.5%) of obtaining the correct answer purely by guessing. To reduce the impact of such randomness, we explored adding an additional reward or weight based on the model's answer confidence. Specifically, we experimented with several approaches to compute confidence scores from the softmax probabilities of the output tokens. However, these confidence-based reweighting strategies did not yield any performance improvements. We analyze that this may be due to the fact that the softmax probability

of the predicted token does not reliably reflect the model's true certainty about the overall answer. In particular, the model may assign high confidence to tokens that are syntactically or semantically unrelated to the actual correctness of the reasoning (e.g., punctuation, or irrelevant words within the output). As a result, the computed "confidence" can be misleading, making it an ineffective signal for reward shaping.

**Importance sampling.** As in our Augmented GRPO, we sample the trajectories on simple examples with weak augmentations, but we use the trajectory to optimize harder exaples with strong augmentations. This might cause misalignment similar to offline reinforcement learning. In this way, we apply importance sampling, which calculates the probability gap between the trajectories for the simple and hard examples as a weight to reweight the reward/advantages. This strategy could not bring improvements. We suspect that although importance sampling is theoretically justified, it may interfere with the core optimization dynamics of GRPO. Specifically, GRPO relies on the relative ranking of trajectories within a group to compute structured advantages. Introducing importance weights—derived from distribution shifts—may distort this internal ranking or inject instability into the reward signals. Additionally, the token-level probability changes caused by visual augmentations can be noisy or poorly calibrated, making the computed importance weights unreliable in practice.

## 5 Conclusion

In this work, we propose a self-supervised framework that leverages inherent image constraints to incentivize multi-image reasoning in VLMs. We identify that the core challenge lies in linking visual cues across images. To address this, we adopt contrastive learning principles and construct image triplets for reinforcement training. To further enhance reasoning, we introduce Augmented GRPO, which samples rollouts from simpler examples and optimizes the model on harder ones. Although trained solely on image comparison tasks, our model generalizes well and achieves strong results across multiple benchmarks.

**Broader Impact.** MiCo explores a self-supervised and reinforcement learning-based approach to improve multi-image reasoning in vision-language models without relying on human-annotated question-answer pairs. By leveraging intrinsic visual constraints, such as consistency across augmented views and differences between similar images, MiCo significantly reduces the need for labor-intensive data curation. This has the potential to democratize the development of reasoning-capable AI systems, making them more accessible in low-resource settings or for underrepresented languages and domains where curated datasets are scarce.

However, as with any powerful vision-language technology, there is a risk of misuse, particularly in applications involving surveillance, misinformation, or unauthorized inference of user intent from visual data. MiCo's improved ability to perform fine-grained comparisons across images could be exploited in privacy-invading scenarios if deployed irresponsibly. To mitigate such risks, we advocate for deploying MiCo in alignment with responsible AI guidelines, ensuring transparency, consent, and clear boundaries in its application domains. In practice, this includes integrating robust sensitive content filtering, restricting deployment in high-stakes or privacy-sensitive scenarios, and establishing human-in-the-loop mechanisms for critical decision-making processes.

**Limitations and future directions.** While our approach supports general reasoning through visual comparisons, it shows limited effectiveness on specialized tasks such as face verification, visual math, and spatial understanding, where structured priors or domain-specific knowledge are required. In future work, we plan to explore more efficient data construction strategies tailored to these domains.

**Acknowledgement.** This work is supported by the National Natural Science Foundation of China (No. 624B2124, 62441615, 62422606).

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
