# OpenReview forum: "MiCo: Multi-image Contrast for Reinforcement Visual Reasoning"
_NeurIPS.cc/2025/Conference — NeurIPS 2025 poster_

### Official Review · Reviewer_Zh8d · 2025-06-30

**Clarity:** 2
**Significance:** 2
**Originality:** 2
**Rating:** 3
**Confidence:** 5

**Summary:**

The paper introduces MiCo, a self-supervised framework to enhance multi-image reasoning in Vision-Language Models (VLMs). Current VLMs struggle with linking fine-grained visual cues across images, especially for tasks requiring spatial alignment, object correspondence, or tracking semantic changes. To address this, MiCo leverages contrastive triplet training and rule-based reinforcement learning. Experiments show MiCo outperforms other SOTA methods on multi-image benchmarks and generalizes to single-image tasks.

**Questions:**

Why do the results reported in Table 3 differ from those originally reported in the papers for MM-Eureka-7B and NoisyRollout-7B?

I would like to know how many samples the authors collected from image edit and video datasets for training, respectively.

**Ethical Concerns:**

["NO or VERY MINOR ethics concerns only"]

**Final Justification:**

The authors' reply has clarified certain aspects of my initial concerns; however, some key issues remain unresolved.

**Limitations:**

yes

**Quality:**

2

**Strengths And Weaknesses:**

**Strengths**

The paper combines self-supervised learning with reinforcement learning, eliminating the dependency on annotated QA pairs. This approach leverages inherent structural constraints for reward calculation.

MiCo achieves SOTA results on VLM2-Bench, surpassing GPT-4o and other advanced VLMs. It can also generalize to diverse tasks, improving multi-image reasoning and even single-image tasks.

The paper conducts comprehensive ablation studies to validate the contribution of each component.

**Weaknesses**

There exists a certain degree of inadequacy in the construction of image triplets. For example, in the lower-left triplet of Figure 2, the first and second images are defined as identical, while the third is defined as different. Yet, compared to the first image, the second image has the person cropped out. While the distinction lies in "the presence/absence of a license plate," I argue that the presence/absence of the person constitutes a more salient difference.

I suspect there may be issues with the authors' SFT setup. Apart from the reported poor performance on the person track, the SFT approach actually underperforms the base model in most cases on both the general and object tracks, exhibiting a significant performance drop. I am curious about how the authors constructed the CoT for SFT. Additionally, I would like to understand why MiCo shows notably lower performance on the person track compared to other reasoning-based models.

The proposed model does not demonstrate a significant improvement over other SOTA methods (e.g., VLAA-Thinker-7B) on the VLM2-Bench. Moreover, the gains on general datasets are also quite limited.

---

> ### Author Rebuttal · Authors · 2025-07-31
>
> ``W1. There exists a certain degree of inadequacy in the construction of image triplets...``
>
> Thank you for the insightful comment. We had similar concerns during development and found cropping to be important both theoretically and empirically. Our reasoning is as follows:
>
> **Cropping mitigates shortcut learning.** Cropping increases the difficulty of the decision, which is **intentional**. Without crop, it is trivial to identify two images as “the same” since their features are exactly identical.Thus, the model may comparing pixel-by-pixel and declaring images “different” upon finding any mismatched pixels. By applying cropping on images labeled as "same," we effectively prevent this and force the model to perform higher-level reasoning to distinguish real content changes rather than relying on superficial visual differences.
>
> **Prompting helps ignore crop-induced differences.** As noted in Line 140, we include instructions like "Regardless of the augmentation" to guide the model away from treating cropping as meaningful variation. This, along with the steady reward curves and performance gains under stronger augmentations (Table 3(c)), suggests the model learns to focus on relevant content. In addition, our cropping is not very aggressive, which crops a region with 0.70-0.95 length of the original image.
>
>
> **Cropping is standard in self-supervised learning.** Contrastive methods like SimCLR, MoCo, and DINO use aggressive cropping to create different views of the same image. These practices show that models can learn to focus on semantic consistency despite visual alterations.
>
> In summary, cropping is a deliberate and validated design choice that increases task difficulty, prevents shortcut learning, and promotes meaningful comparison.  Our experimental results in Table 2(c) also verifies its effectiveness.
>
> ---
>
> ``W2. I suspect there may be issues with the authors' SFT setup. ...I am curious about how the authors constructed the CoT for SFT. ``
>
> **Details of the SFT setup.**  As noted in Lines 207–208, we apply SFT on our contrastive dataset to directly predict binary labels (“same” / “different”) without using CoTs, and this model fails to generalize to the more diverse tasks in VLM2-Bench. In contrast, using the same supervision, our RL-based method shows much stronger generalization. This SFT setting differs from a typical “CoT cold-start” scenario, which we will clarify in the revision.
>
> **SFT on CoTs.** Following your suggestion, we also conducted SFT with CoT-labeled samples in the third row of the following table. We prompted Qwen2.5-VL-7B to generate reasoning CoTs for contrastive triplets and used rejection sampling to retain only CoTs leading to correct answers. This brings moderate improvements over answer-only SFT, but still lags behind our RL-based solution. The results highlight the challenge of generalizing from binary classification to open-ended reasoning, and further justify the use of RL.
>
>
>
> | Model | General  | Object | Person |
> | -------- | -------- | -------- | -------- |
> | Qwen2.5VL-7B    |   39.64  | 	53.53  | 63.44  |
> | SFT(answer only) | 42.90 | 51.15 | 55.98 |
> | SFT(with CoT) | 46.13	| 57.26 |	54.52 |
> | RL(with CoT)    | 62.13	|  65.53	|  57.18 |
>
> ---
>
> ``W2. Additionally, I would like to understand why MiCo shows notably lower performance on the person track compared to other reasoning-based models.``
>
>
>
> | Model | General  | Object | Person |
> | -------- | -------- | -------- | -------- |
> | Qwen2.5VL-7B    |   39.64  | 	53.53  | 63.44  |
> | Qwen2.5VL-7B + raw CoT      |  43.08	| 50.98	| 48.52|
> | MM-Eureka    | 51.31  | 	60.20 | 58.00 |
> | NoisyRollout    | 42.38 | 	49.55 | 	54.33 |
> | VLAA-Thinker    | 55.26 | 	62.86 | 	56.81 |
> | ThinkLite    | 43.51 | 	62.52 | 	56.87 |
> | MiCo(Ours)    | 62.13	|  65.53	|  57.18 |
> | MiCo(Ours) + Person Specific    | 61.86	|  63.90	|  65.26 |
>
> **Our results on the person-centric track are competitive.** We respectfully clarify that our performance is not notably lower than other reasoning models. Our method achieves 57.18 on the person track—slightly below MM-Eureka but outperforming all other solutions—demonstrating strong competitiveness.
>
>
>
> **Face verification differs significantly from other tasks.** Person-track focuses on identity-related questions. Unlike object comparison or spatial reasoning, face verification and Re-ID require fine-grained, abstract perception. While our method improves tasks involving color, texture, or spatial cues, these contrastive signals are less effective—and sometimes distracting—for identifying whether two faces belong to the same person.
>
>
>
> **General CoTs degrade person-track performance.** We observe that all reasoning-based models show reduced performance on person-centric tasks compared to the base model (63.44). In particular, directly adding a CoT prompt leads to a significant drop (from 63.44 to 48.52). We argue that person tasks rely more on direct visual representation comparison than on text-based reasoning.
>
>
> **Could be improved by task-specific designs.**  While the performance on person-centric tasks could be improved by incorporating task-specific data (as we add LFW dataset in the last row), our current focus is on developing a general and broadly applicable framework. We intentionally avoid tailoring the data or training strategy to individual task types, in order to preserve the generality and transparency of our method. Instead, we provide an honest analysis of the limitations observed in certain task categories, which we believe offers more meaningful insights for future research.
>
>
> ---
>
>
>
> `` W3. The proposed model does not demonstrate a significant improvement over other SOTA methods (e.g., VLAA-Thinker-7B) on the VLM2-Bench. Moreover, the gains on general datasets are also quite limited.``
>
> **Cost for training data construction.** We would like to emphasize that our main insight lies in leveraging self-supervised signals to train RL models. We only utilize existing video and image editing datasets without any additional manual processing. In contrast, the main contribution of other works are the construction of a training dataset with complicated pipeline(sample collection, labling, etc)  and even with human-verified answers, which are significantly more costly.
>
> Even without any human annotation or verification, our method still achieves state-of-the-art performance with a margin of 3%. Imagine extending their data construction pipeline to cover fine-grained comparisons across multiple images—the cost would increase substantially (as labeling multi-image data is far more complex), and there is no guarantee that the resulting performance would surpass ours.
>
> **They also include "multiple images".** Other worlk like MM-Eureka dataset also contains examples involving multiple sub-figures, such as comparing different diagrams in a long image or understanding three-view technical drawings. Although these are concatenated into a single image input, such compositions still help the VLM learn to link visual cues across different images.
>
> **The performance is already high.** Compared with our baseline (Qwen2.5-VL-7B-CoT), we have already achieved a 12.93% improvement, even surpassing GPT-4o. When performance is already high, further improvements become quantitatively more difficult to obtain.
>
> **We get higer results on single-image task without training.** We only train our model on multiple images comparing task, however, our method also brings improvements on many single image understanding benchmarks, and even achieves higher performance then MM-Eureka on HallusionBench, MMMU, etc., as shown in Table 3. We think the improvements on single image benchmarks are already promising considering we only leverage multi-image self-supervised rewards.
>
> **Concurrent works.**  Additionally, please notice that these works are concurrent works. Eventhough,we add them to discussion and surpasses them with a clear margin(3%)
>
>
>
>
>
> ---
>
> ``Q1. Why do the results reported in Table 3 differ from those originally reported in the papers for MM-Eureka-7B and NoisyRollout-7B?``
>
> Thank you for your careful review. As noted in Lines 176–179, we follow the default hyperparameters of Qwen2.5-VL and use the VLMEvalKit codebase for evaluation.
>
> Minor discrepancies may arise because different methods may have modified question prompts or applied custom answer parsing strategies. In practice, we found it difficult to exactly reproduce some reported results due to such variations and potential missing implementation details.
>
> Still, all methods are based on Qwen2.5-VL-7B, and we strictly follow their system prompt formats. We believe our evaluation is fair and consistent under the same settings.
>
>
>
> ---
>
> ``Q2. I would like to know how many samples the authors collected from image edit and video datasets for training, respectively.``
>
> We use a total of 10,000 training samples, consisting of 5,000 from image editing datasets and 5,000 from video datasets. This data scale is in line with prior VLM reasoning works—for example, Vision-R1 (10K), NoisyRollout (6.4K), ThinkLite-7B-VL (11K), and MM-Eureka (15K)—ensuring a fair basis for comparison.

---

> > ### Author Response · Authors · 2025-08-04
> >
> > Dear Reviewer Zh8d,
> >
> > We sincerely appreciate the time and effort you have spent reviewing our submission. We hope that our rebuttal has addressed your concerns adequately.
> >
> > If you have a chance, we would be very grateful for any further feedback you might have, or any clarification you may need from our side. We would be happy to engage in additional discussion if that would be helpful.
> >
> > Thank you again for your thoughtful review.

---

> > ### Author Response · Authors · 2025-08-05
> >
> > Dear Reviewer Zh8d,
> >
> > Thank you once again for your valuable feedback on our paper. In response to your comments, we have included additional experiments and clarifications to better address your concerns.
> >
> > As the author-reviewer discussion period is drawing to a close, we would greatly appreciate it if you could let us know whether our responses have sufficiently addressed your questions.
> >
> > We are sincerely grateful for your time and thoughtful consideration.

---

> > ### Comment · Reviewer_Zh8d · 2025-08-06
> >
> > Thank you for the author's rebuttal. However, I am still unsatisfied with the responses and have the following remaining concerns:
> >
> > W1: Cropping is indeed a classic technique in self-supervised learning and contrastive learning for constructing positive samples. However, when constructing negative samples, it is common practice to ensure that the discrepancy between the negative sample and the anchor sample is greater than that between the anchor and positive samples. For instance, different samples within the same batch (e.g., SimCLR) or samples from different categories are typically used as negatives. In the author's scenario, however, taking the bottom-left corner of Figure 2 as an example, the discrepancy between the positive sample and the anchor is a few people, while the discrepancy between the negative sample and the anchor is merely a license plate. This construction method appears to result in a smaller discrepancy between the positive sample and the anchor compared to that between the negative sample and the anchor.
> >
> > W2: Why does SFT outperform the base model in the General subtask but underperform in the Object and Person subtasks? Could the author clarify the number of training steps and the dataset size used for SFT?
> >
> > W2: Moreover, in Table 1's Person subtask, the cpr and cnt metrics are significantly worse than those of other reasoning models, which is unsatisfactory.
> >
> > W3: I acknowledge the author's explanation regarding dataset construction. However, I cannot agree with the claimed performance improvements. First, the task design itself is tailored for fine-grained visual reasoning, and given that a comparable number of samples were used as in other reasoning models, the improvement in relevant tasks is not convincing. The author claims significant improvements over Qwen2.5-VL-7B-CoT, but in fact, the CoT version exhibits a noticeable performance drop compared to the base version, making this claim questionable. As for the comparison with GPT-4o, it is not a reasoning-specific model. A more appropriate comparison should be made against models like the O1 or O3 series (no additional experiments are required at this stage; I merely raise this point to rebut the author's argument).

---

> ### Author Response · Authors · 2025-08-06
>
> Thank you for your thoughtful comments. We provide the following clarifications and hope they address your concerns.
>
>
>  ``W1: Concerns about cropping.``
>
> **We instruct the model to ignore cropping through explicit prompting.** As explained in the rebuttal, we provide prompts such as "Regardless of the augmentation" or "Regardless of the cropping/resizing" to guide the model to focus on meaningful changes. Moreover, in the model’s reasoning chains, we observe generated phrases like "besides the viewpoint changes...", which suggests the model understands and distinguishes cropping-related transformations.
>
> **Real image changes are more salient than cropping effects.** Our cropping ratio is within 0.7–0.95, which is relatively conservative. We kindly remind the reviewer that in real-world tasks such as image editing or video understanding, the actual changes are often more prominent than simple cropping. While Table 2 includes some challenging cases to illustrate the effectiveness of augmentations, we plan to include more representative examples in the revision to better reflect the overall data distribution.
>
> ---
>
> ``W2: Why does SFT outperform the base model in the General subtask but underperform in the Object and Person subtasks? Could the author clarify the number of training steps and the dataset size used for SFT?``
>
> Thank you for your careful review.  While reinforcement learning (RL) generally improves generalization, it still performs better when the training data distribution aligns with the target task. The General Track poses questions (e.g., finding image differences) that are more similar to our training settings. Even though we cannot simulate identity-level changes, our model generalizes reasonably well to the Object Track by focusing on fine-grained visual details. However, the Person Track focuses on human ID recognition, which is highly domain-specific, and our model struggles to generalize in this area.
>
> Breakdown of the Tracks:
>
> General Track: Comparing fine-grained differences or connections between visually similar images.
>
> Object Track: Identifying object-level identity among appearance-similar objects.
>
> Person Track: Identifying human identity (person re-ID).
>
> Regarding the training setup: we strictly follow the RL training protocol by constructing 10K self-supervised samples and training for 600 steps.
>
> ---
>
> ``W2: Moreover, in Table 1's Person subtask, the cpr and cnt metrics are significantly worse than those of other reasoning models, which is unsatisfactory.``
>
> Thank you for your detailed question. We explain the reason why we achieve high grp (grouping) performance but relatively unsatisfactory cpr (comparison) and cnt (counting) scores.
>
> We observe that MiCo is good at finding fine-grained differences among images; however, these differences may influence person-ID judgment. Since the model tends to focus on differences in appearance, clothing, and expressions, it has a higher tendency to identify images as belonging to different persons. In contrast, other concurrent works fail to capture such fine-grained differences, which leads to better results in cpr (comparing whether two images show the same person) and cnt (counting the number of different persons).
>
> However, in the grp task (grouping people by their identities), MiCo can also identify more significant differences such as age, gender, or race, rather than just clothing or expressions. In this case, MiCo yields better results.
>
> Overall, this is because person-ID is a specific tasks, the reasoning process should be different from the CoT trained on general image comparison, this issue could be simply addressed by adding person-ID data in practical usage. As for search works, we make it transparent for disscussion.
>
> ``W3: I acknowledge the author's explanation regarding dataset construction. However, I cannot agree with the claimed performance improvements. ``
>
> **The performance is already significant.** Compared with Qwen2.5-VL (without CoT), our method achieves a 6.2% gain on VLM2-Bench. We also report +2.1%, +1.8%, and +2.4% gains on MuirBench, BLINK, and MMMU, respectively. For general vision tasks—where performance tends to saturate—these improvements are already meaningful.
>
> **We only use self-supervision.** We would like to emphasize that our method requires no human annotation. It provides a new perspective to the community: that leveraging the inherent constraints within image/video pairs can yield steady improvements. We believe this is a valuable and scalable direction for future research

---

> > ### Author Response · Authors · 2025-08-07
> >
> > Dear Reviewer,
> >
> > Thank you again for your time and thoughtful feedback on our submission. We have provided detailed responses to your comments, and we truly appreciate the opportunity to address your concerns.
> >
> > As the discussion period is nearing its end, we would be very grateful if you could take a moment to revisit our responses and let us know whether they have resolved your questions, or if there are remaining points you would like us to clarify further.
> >
> > We are happy to engage in further discussion if needed. Thank you again for your time and consideration.

---

> > ### Author Response · Authors · 2025-08-08
> >
> > Dear reviewer,  we would like to add some explanations on our **performance improvements**.
> >
> > First, we emphasize again that we give 6% improvements on our main task compared with baseline, and  +3% with concurrent SoTAs.  Then, we give  +2.1%, +1.8%, and +2.4% gains on MuirBench, BLINK, and MMIU, respectively.
> >
> > **The major misunderstanding** might that we are not that dominant for other multi-image benchmarks compared with concurrent SoTAs. However, the reason is that MuirBench, BLINK, and MMIU are not that suitable for multi-image understanding. There are many questions that let the model tackle each images independently.   We first extract some "real multi-image" tasks that ask the model to compare the fine-grained visual cues among images as follows, and we observe that MiCo brings huge improvements over competitors.
> >
> > | Model       | Semantic_Correspondence | Spatial_Relation | Visual_Similarity | Visual_Retrieval | Geographic_Understanding |
> > |-------------|-------------------------|------------------|-------------------|------------------|--------------------------|
> > | MM-Eureka    | 0.33             | 0.82     | 0.77     | 0.57       | 0.44                     |
> > | MiCo      | 0.46  **(+13%)**            | 0.90 **(+8%)**     | 0.85 **(+8%)**      | 0.71 **(+14%)**       | 0.53 **(+9%)**                     |
> >
> >
> > **However, these benchmark also contains many "fake multi-image" questions**, which makes our overall improvements not that dominant. For example, these questions count for over 50% in MuirBench, similar issues also happens in BLINK and MMIU. In this case, we emphasize the importance on the performance on VLM2-Bench, which is the mostly aligned with our motivation.
> >
> > | Task Name              | Ratio   | Example                                                                                                  |
> > |------------------------|---------|----------------------------------------------------------------------------------------------------------|
> > | Image text alignment   | 17.80%  | Which picture below better fits the description: ...                                                     |
> > | Diagram Understanding  | 15.30%  | Which shape has an area of 6 square units? The shapes are made of unit squares.                  |
> > | Counting               | 9%      | How many other garments besides a complete mitten pair are shown in each image?                         |
> > | Scene Understanding    | 7.20%   | what colors are painted along the curb in the given images? |
> > | Cartoon Understanding  | 3%      | What is the main content of this comic strip?                            |

---

> > > ### Author Response · Authors · 2025-08-09
> > >
> > > Dear Reviewer,
> > >
> > > We truly appreciate the time and effort you have already dedicated to reviewing our submission and engaging with our responses. Your feedback has been very helpful in improving the clarity and quality of our work.
> > >
> > > As the discussion phase is approaching its conclusion, we wanted to kindly check in and see if there are any remaining concerns or questions you would like us to address. We would, of course, be happy to clarify or expand on any points if needed.
> > >
> > > Thank you again for your thoughtful contributions, we greatly value your input and the opportunity to further improve our manuscript based on your insights.

---

### Official Review · Reviewer_jEYQ · 2025-07-02

**Clarity:** 3
**Significance:** 2
**Originality:** 3
**Rating:** 4
**Confidence:** 4

**Summary:**

The paper presents a post-training framework named MiCo (Multi-image Contrast), to enhance the multi-image reasoning capabilities of Vision-Language Models (VLMs) through reinforcement learning (RL). The major contributions of this work are its task definition for training and the automated pipeline to construct labeled training data.

MiCo trains a VLM to find two augmented views of the same image in a triplet of three similar images (views). The model is optimized to generate chain-of-thought (CoT) reasoning to determine similarities/differences among the triplet. The training pipeline, including prompt format and reward design, is basically borrowed from the DeepSeek-R1 and its GRPO algorithm. Besides, the paper introduces a rollout augmentation strategy that uses weakly augmented views to rollout and strong augmented views to update the model.

MiCo can constructs labeled data for RL training, without the need for human annotations. It constructs image triplets consisting of two augmented views of the same image and a third similar but distinct image, sourced from video frames (e.g., VidGen-1M) and image editing datasets (e.g., OmniEdit).

Experiments show that MiCo, without relying on manual annotated training data, generalizes effectively to diverse multi-image reasoning benchmarks (e.g., VLM2-Bench, MuirBench) and achieves state-of-the-art results.

**Questions:**

1. The paper does not discuss the data scaling for RL training. What is the minimum amount of data required to train the model, and does the model performance continue to improve with more data and more training steps?
2. In Section 3.3, the paper mentions "To increase the diversity and balance the difficulties of questions, besides the image triplet, we also
142 construct image pairs and design the corresponding prompts for comparing two images." What is the detail of this process? How does it affect the model performance?
3. Do the training data include samples with triple views of the same image? In other words, can the labels be "TTT" or "FFF", or must have twice "T" and once "F"? Also, can the training data include samples with only two views? How does these modifications affect the model performance?
4. In table 1, why is Qwen2.5-VL-7B-CoT marked as non-reasoning model?
5. Does the Prompt Diversity in Table 2 (e) mean using variants of reasoning prompt (system prompt) or variants of user question template, or both (that are defined in Section 3.3)?

**Ethical Concerns:**

["NO or VERY MINOR ethics concerns only"]

**Final Justification:**

The paper provides simple ideas and an effective method to improve MLLM's multi-image reasoning abilities. The experimental results and conclusions of the paper are convincing. Thus, I believe the paper is worthy of acceptance.

**Limitations:**

yes

**Quality:**

3

**Strengths And Weaknesses:**

**Strengths:**
- The training paradigm of MiCo is simple yet effective. The training task of comparing image triplets is conceptually straightforward and simple. Yet, it is effective in enhancing the multi-image reasoning capabilities of VLMs.
- MiCo can automatically constructs training QA pairs with image intrinsic constraints, significantly reducing dependency on labor-intensive annotations. Also, it essentially implements a self-supervised learning paradigm.
- It proposes a rollout augmentation strategy which uses weakly augmented views to rollout and use strong augmented views to update the model. This strategy is shown to be effective in improving the model's reasoning capabilities.
- The trained model demonstrates strong multi-image reasoning capabilities and also generalizes well to general visual language tasks.
- The ablation experiments are comprehensive and insightful.


**Weaknesses:**
- The framework only applies to the Qwen2.5-VL-7B model, which rises concerns about its generalizability to other VLMs, including larger models as well as other model series (e.g., LLaMA, Mistral).
- The paper lacks discussion on the data scaling for RL training. It is unclear the minimum amount of data required to train the model, and whether the model performance continues to improve with more data and more training steps.
- The framework relies heavily on high-quality contrastive pairs (from videos or editing datasets). Its performance may degrade with less structured or noisier data sources. This also poses a challenge for the framework if we want to scale the training data.
- The framework may degrade model's performance on person-centric tasks. The paper does not provide sufficient analysis on this aspect.

---

> ### Author Rebuttal · Authors · 2025-07-31
>
> ``W1. The framework only applies to the Qwen2.5-VL-7B model, which rises concerns about its generalizability to other VLMs, including larger models as well as other model series (e.g., LLaMA, Mistral).``
>
> **Previous works also use Qwen2.5-VL.** Thank you for your constructive suggestion. We would like to clarify that nearly all concurrent works on RL-based CoT—such as MM-Eureka, NoisyRollout, and ThinkLite-VL—conduct experiments on Qwen2.5-VL-7B. We follow this common setting to ensure fair comparison and consistency in base configurations.
>
> **Results on more models.** In response to your suggestion, we also evaluate our method on additional base models. Since our framework requires a vision-language model (VLM), pure language models (e.g., LLaMA, Mistral) are not applicable. Therefore, we extend our experiments to different model sizes of Qwen2.5-VL as well as MIMO-VL-7B, and evaluate them on VLM2-Bench.
>
> In all settings, the base models are prompted to generate CoTs before producing the final answers.Our method consistently brings improvements. The gains are more significant when the baseline is relatively weak, but we still observe clear improvements even with strong baselines.
>
>
> | Model | General  | Object | Person | Overall |
> | -------- | -------- | -------- | -------- | -------- |
> | Qwen2.5VL-3B     | 20.89 | 47.94 | 35.95 | 34.93 |
> | Qwen2.5VL-3B + Ours     | 34.57 |	57.72	| 44.66 |45.65 |
> | Qwen2.5VL-32B     | 63.29 | 71.81 | 65.16 | 66.75 |
> | Qwen2.5VL-32B + Ours     |67.42 | 73.21 | 67.22 | 69.28 |
> | MIMO-VL-7B     | 71.26 |68.99 | 59.52 |66.59 |
> | MIMO-VL-7B + Ours     | 74.38 | 70.96 | 58.83 | 68.06  |
>
> * Please note that training 32B models requires significant computational resources. Therefore, we used a smaller group size (4) and trained for fewer steps (100 steps).
>
> ---
>
>
> ``W2. The paper lacks discussion on the data scaling for RL training....``
>
> Thank you for your suggestion. We add ablations for data scaling. In the main paper, we use 10K samples for training to ensure fair comparisons, in line with recent RL works: Vision-R1 (10K), NoisyRollout (6.4K), ThinkLite-7B-VL (11K), and MM-Eureka (15K).
>
> We report ablation results on VLM2-Bench with varying data scales:
>
> | Num. Data | 0  | 100  | 500 | 5,000 | 10,000 | 50,000  | 100,000 |
> | -------- | -------- | -------- | -------- | -------- |-------- |-------- |-------- |
> | Performance  | 52.20  |  55.02  | 56.25   | 58.89  | 61.61  | 62.53  | 58.32 (sometimes collapse) |
>
> We provide the following observations:
> * **Small-scale data is also effective.** Even with only 100 training samples, we observe improvements over the baseline. This reflects a typical property of RL—its effectiveness with limited supervision. In addition, our augmentation strategy increases sample diversity, further enhancing learning efficiency.
> * **Scaling law under moderate data regimes.** Increasing the amount of training data within a moderate range leads to steady performance gains, with results improving from 55.02 to 62.53 as the data size increases from 100 to 50K.
> * **Saturation and model collapse at large scale.** When scaling to 100K samples (and training for more steps), performance begins to drop, and sometimes we meet "model collapse". We already include KL regularization to stabilize training. We hypothesize that the model may overfit to the task-specific data and lose general capabilities. Given that our task scope is relatively narrow, a more suitable strategy may be interleaved training (as in DeepSeek-R1), which alternates between rule-based RL for specific tasks and supervised fine-tuning (SFT) for general capabilities.
>
>
>
> ---
>
>
> ``W3. The framework relies heavily on high-quality contrastive pairs (from videos or editing datasets)...``
>
> We would like to respectfully clarify that a core motivation of our framework is to eliminate the need for costly data preparation. In fact, this is a key advantage of our approach over prior methods, which often rely on expensive and carefully curated datasets.
>
> Our solution does not depend on any specific data source—it can leverage arbitrary video sources or image editing datasets, as long as we can obtain pairs of similar but different images. As shown in Table 2(b), using different editing datasets does not significantly affect the final performance, indicating the robustness of our method to data variation.
>
> ---
>
> ``W4. The framework may degrade model's performance on person-centric tasks. The paper does not provide sufficient analysis on this aspect.``
>
> | Model | General  | Object | Person |
> | -------- | -------- | -------- | -------- |
> | Qwen2.5VL-7B    |   39.64  | 	53.53  | 63.44  |
> | Qwen2.5VL-7B + raw CoT      |  43.08	| 50.98	| 48.52|
> | MM-Eureka    | 51.31  | 	60.20 | 58.00 |
> | NoisyRollout    | 42.38 | 	49.55 | 	54.33 |
> | VLAA-Thinker    | 55.26 | 	62.86 | 	56.81 |
> | ThinkLite    | 43.51 | 	62.52 | 	56.87 |
> | MiCo(Ours)    | 62.13	|  65.53	|  57.18 |
> | MiCo(Ours) + Person Specific    | 61.86	|  63.90	|  65.26 |
>
> **Face verification differs significantly from other tasks.** Person-track focus on the questions related to human ID. However, face verification / Re-ID requires more abstract and fine-grained perception compared to other tasks that benefit from visual cue linking (e.g., object comparison or spatial reasoning). While our method improves tasks involving color, texture, or relative position analysis, these contrastive visual cues may not be helpful for determining identity between two faces—and can even introduce distractions.
>
> **Our results on the person-centric track are competitive.** We report per-track performance across models. Our method achieves 57.18 on the person track, slightly below MM-Eureka but outperforming all other baselines, demonstrating competitive performance.
>
> **CoT generally degrades person-track performance.** We observe that all reasoning-based models show reduced performance on person-centric tasks compared to the base model (63.44). In particular, directly adding a CoT prompt leads to a significant drop (from 63.44 to 48.52). We argue that tasks such as face verification are less suitable for CoT-style reasoning, as they rely more on direct visual representation comparison than on text-based reasoning.
>
>
> **Could be improved by task-specific designs.**  While the performance on person-centric tasks could be improved by incorporating task-specific data (as we add LFW dataset in the last row), our current focus is on developing a general and broadly applicable framework. We intentionally avoid tailoring the data or training strategy to individual task types, in order to preserve the generality and transparency of our method. Instead, we provide an honest analysis of the limitations observed in certain task categories, which we believe offers more meaningful insights for future research.
>
> ---
>
>
> ``Q1. The paper does not discuss the data scaling for RL training...``
>
> Please see the answer in W2.
>
> ---
>
>
> ``Q2. In Section 3.3, the paper mentions "To increase the diversity and balance the difficulties of questions, besides the image triplet, we also construct image pairs and design the corresponding prompts for comparing two images." What is the detail of this process? How does it affect the model performance?``
>
> Literally, we construct a portion of samples using only two images and ask the model whether they are different. The answer is typically a simple True/False. Since many real-world multi-image understanding tasks essentially involve comparing two images (e.g., finding differences, identifying changes, verifying consistency), including such samples improves the diversity and balance of the training data. As shown in Table 2(d), this design leads to measurable performance improvements.
>
> ---
>
> ``Q3. Do the training data include samples with triple views of the same image? In other words, can the labels be "TTT" or "FFF", or must have twice "T" and once "F"? Also, can the training data include samples with only two views? How does these modifications affect the model performance?``
>
> Currently, we support TTT/FFF configurations in the video data, as we are able to sample three distinct frames from the same or different scenes. However, image editing datasets typically provide only two different images, so in those cases, we can only construct triplets where all three images are the same (TTT) or where a third different image is inserted.
>
> This configuration does not significantly affect overall performance. We conduct ablation studies using different variations of the video dataset to analyze this aspect in more detail, as shown below:
>
> |          | General  | Object   | Person  |
> | -------- | -------- | -------- |-------- |
> | Video Data | 60.29 | 64.50 | 55.68 |
> | Video Data(w/o 3 different) | 60.12 | 65.10 | 55.05 |
> | Edit + Video | 62.13 | 65.53 | 57.18 |
> | Edit + Video(w/o 3 different) | 63.31 | 64.92 | 56.50 |
>
>
> ---
>
>
> ``Q4. In table 1, why is Qwen2.5-VL-7B-CoT marked as non-reasoning model?``
>
> Qwen2.5-VL-7B is not trained with reinforcement learning. We simply prompt it to perform reasoning by providing a reasoning-style instruction, such as: "First output the thinking process in <think></think>, and give the final answer in <answer></answer> tags."
> Thus, it does not undergo reasoning-specific training, and we categorize it as a non-reasoning model. We will clarify this definition in the revised version of the paper.
>
> ---
>
>
> ``Q5. Does the Prompt Diversity in Table 2 (e) mean using variants of reasoning prompt (system prompt) or variants of user question template, or both (that are defined in Section 3.3)?``
>
> Prompt diversity in Table 2(e) refers to using different user question templates. Specifically, we use GPT-4o to rewrite multiple question templates with similar meanings to increase variability. We keep the system prompt fixed, following common practice, as consistency is important during inference and evaluation.

---

> > ### Comment · Reviewer_jEYQ · 2025-08-03
> >
> > Thank you to the Authors for your patient rebuttal; my concerns have now largely been addressed.

---

> > > ### Author Response · Authors · 2025-08-04
> > >
> > > We appreciate your valuable comments. We would further polish this paper according to your suggestions.

---

### Official Review · Reviewer_A4x6 · 2025-07-03

**Clarity:** 4
**Significance:** 3
**Originality:** 3
**Rating:** 5
**Confidence:** 3

**Summary:**

The authors propose a method to enable vision-language models (VLMs) to perform multi-image reasoning. They begin by demonstrating that existing VLMs struggle with robust visual comparison. To address this, they construct training triplets consisting of two augmentations of the same image and a third image that is visually similar but semantically different. Using these triplets, they apply an augmented Group-Relative Policy Optimization (GRPO) approach to refine reasoning trajectories. Experimental results show that the method achieves competitive performance across multiple benchmarks.

**Questions:**

It would be interesting to see more failure cases and corresponding qualitative results. Could the authors also elaborate on the scale of the training data and whether they were manually validated?

**Ethical Concerns:**

["NO or VERY MINOR ethics concerns only"]

**Final Justification:**

Given the rebuttal that addresses my remaining concerns, I retain my positive score.

**Limitations:**

Yes

**Quality:**

3

**Strengths And Weaknesses:**

### Strengths

- The idea to perform augmented GRPO on image triplets is novel, addresses a well-defined problem (multi-image understanding and reasoning), and does not require manual annotations.

- The authors demonstrate significant improvements over the baseline, showcasing the strength of the method.

- The authors also conduct extensive ablation studies.

### Weaknesses

- The use of synthetic data and augmentations during training could lead to shortcut learning.

- For reproducibility, it would be important to include more details about how the editing pipelines, such as OmniEdit, were used to construct the data (e.g., what types of prompts were used).

---

> ### Author Rebuttal · Authors · 2025-07-30
>
> Thank you for acknowledging our paper and for your constructive comments. We will further polish our manuscript according to your suggestions.
>
> ---
>
> ``W1. The use of synthetic data and augmentations during training could lead to shortcut learning.``
>
> **Augmentation could alleviate shortcut learning.** We would like to clarify that the purpose of adding augmentations is not to create "pairs with different images," but rather to make "pairs with the same image" more challenging. Without augmentation, it would be trivial for the model to identify identical images, as their visual tokens are exactly the same. The model could compare two features pixel by pixel, once they find one different pixel, they could conclude they are different images. This may lead to shortcut learning, where the model fails to compare finer details. With augmentations, however, the model can no longer rely on pixel-by-pixel similarity to match images, and is instead forced to attend to semantic content and make the CoT (Chain-of-Thought) reasoning more effective. The results in Table 2(c) demonstrate the importance of augmentation.
>
> **We also incorporate real data from videos.** To mitigate potential biases from synthetic differences, we additionally include video frames, which naturally capture real-world transformations. As shown in Table 2(b), the combination of video data and editing data yields the best performance.
>
> **Experiment results support the effectiveness of synthetic data and aug.** While we acknowledge that synthetic differences may not perfectly align with real-world transformations, reinforcement learning is known for its strong generalization capability. Our experimental results confirm the effectiveness of our approach—achieving state-of-the-art performance on VLM2-Bench and consistent improvements on other benchmarks. Furthermore, Table 2(b) shows that incorporating editing data leads to significant performance gains.
>
> ---
>
> ``W2. For reproducibility, it would be important to include more details about how the editing pipelines, such as OmniEdit, were used to construct the data (e.g., what types of prompts were used).``
>
> **We use existing datasets.** We would like to clarify that we directly use an existing image editing dataset (OmniEdit), where before- and after-editing image pairs are already provided. This configuration is described in Line 169, and we also perform an ablation study on different editing datasets in Lines 214–218 and Table 2(b). Therefore, our method does not involve constructing editing pipelines, generating prompts, or other manual processing.
>
> **Results on self-constructed data.**  Although we primarily use existing datasets in the main paper, we also follow your suggestion and construct editing pairs ourselves.
> Specifically, we use OmniEdit to obtain the original images, prompt Qwen2.5-VL-7B to imagine editing instructions (covering basic editing types such as adding, removing, attribute modification, pose change, etc.), and then use Flux-Kontext to generate the edited images. This pipeline yields comparable or even better results.
>
> | Data | General | Object | Person |
> |--------|--------|--------|--------|
> | OmniEdit | 61.23 | 65.33 | 56.35 |
> | UltraEdit |  60.88 | 64.33 | 55.27 |
> | Our Constructed |  61.46 | 65.51 | 56.28 |
>
> **Open-source for reproducibility.**  To enhance reproducibility, we will open-source the training code and data upon acceptance.
>
> ---
>
> ``Q1. It would be interesting to see more failure cases and corresponding qualitative results. Could the authors also elaborate on the scale of the training data and whether they were manually validated?
> ``
> Thank you for your constructive suggestions, we would add more qualitative analysis for the failura cases.
>
> **Failure cases.**  We have indeed observed several interesting failure cases caused by CoT overthinking. For example, when asked, “Are there any objects in image1 but not in image2?”, the model sometimes responds: “... There is a boy in image1 who disappears in image2. However, a human is not an object, so there are no objects...” This kind of reasoning reflects the model’s tendency to over-interpret the question. We will include a more detailed analysis of such cases in the revised version. Although such cases are relatively infrequent, addressing them may require providing the model with clearer explanations of the questions and more precise definitions of what is expected in the answer.
>
>
> **Training data.** We currently use 10,000 samples (5K from editing data and 5K from video data). This scale is comparable to other VLM reasoning models, such as Vision-R1 (10K), NoisyRollout (6.4K), ThinkLite-7B-VL (11K), and MM-Eureka (15K).
>
> Our data does not require manual validation, which we consider an advantage. Since we aim to explore self-supervised signals from raw images and videos, we inherently know whether two images are the same or different based on how they are constructed. As a result, our final answers are automatically correct without any human annotation.

---

> > ### Comment · Reviewer_A4x6 · 2025-08-07
> >
> > Thank you for the detailed rebuttal. The rebuttal has fully addressed my concerns and I retain my positive score.

---

### Official Review · Reviewer_sVvm · 2025-07-03

**Clarity:** 2
**Significance:** 2
**Originality:** 2
**Rating:** 3
**Confidence:** 5

**Summary:**

MiCo proposes a multi-image contrastive learning framework to train visual language models for cross-image reasoning by constructing image triplets (two enhanced versions of the same image + a similar but different image). The model is prompted to generate reasoning procedures to compare these images and is optimized through rule-based reinforcement learning without the need for manually annotated question-answer pairs.

**Questions:**

refer the weaknesses. I will consider increase my rate within the rebuttal.

**Ethical Concerns:**

["NO or VERY MINOR ethics concerns only"]

**Final Justification:**

Thank you for the author's response. After careful reading, I still hold the following views, and I believe this paper is too mundane to provide effective insights. Therefore, I maintain my original score, and I think a score of 3 is very appropriate for this article.

Although the authors list the differences from NoisyRollout, they must acknowledge that they are still very similar. For example, crop, resize, and noise are all common techniques in image augmentation, and this minor difference is hardly convincing.

The improvements on other multi-image benchmarks chosen by the authors are too marginal, with significant improvements only on the benchmark established by the authors themselves. This raises doubts about the effectiveness of the method.

"We only utilize existing video and image editing datasets without any additional manual processing. In contrast, the main contribution of MM-Eureka is the construction of a training dataset that 'features diverse knowledge domains with human-verified answers and solution processes.'" This statement is not at all an explanation for why their method only shows marginal improvement over Eureka. If the authors believe that open-source datasets are of poor quality, they should collect high-quality data, such as a multi-image version of MMK12, which would make a greater contribution.

**Limitations:**

Yes

**Quality:**

2

**Strengths And Weaknesses:**

Strengths:
1. It uses the intrinsic constraints of images as supervisory signals to construct image triplets that are visually similar but have subtle differences, thus avoiding the high cost of manually constructing QA pairs. And Augmented GRPO, which uses weakly enhanced sampling trajectories and optimizes the strategy on strong enhancement.



Weaknesses:
1. The method lacks necessary insights, similar to the multi-image version of NoisyRollout
2. It is necessary to evaluate on more benchmarks to illustrate the superiority of the method and model, such as MuirBench, MMIU
3. Typo： LVAA-Thinking
4. In a sense, I think the results in Table 1 cannot explain the superiority of Mico. For example, Eureka is only trained in a single-image scenario, but it is only 3 points behind Mico, which is trained on multiple images.

---

> ### Author Rebuttal · Authors · 2025-07-30
>
> ``W1: The method lacks necessary insights, similar to the multi-image version of NoisyRollout.``
>
> We would like to warmly clarify that our Augmented-GRPO is fundamentally different from NoisyRollout. We are aware that the fact “we both use image augmentation” might make us **appear similar**, which is why we have included a detailed explanation in Lines 29–42 of the appendix.
>
> **Here we emphasize the core differences.** NoisyRollout uses image augmentation (primarily Gaussian noise) to increase the diversity of sampled CoTs, and leverages these more diverse CoTs to optimize the original, non-augmented prompts.
>
> In contrast, we apply more aggressive augmentations (e.g., cropping, resizing) and optimize the strongly augmented samples directly. Our motivation is to use easier samples (with weak augmentation) to obtain better CoTs, and then use these CoTs to optimize the harder samples (with strong augmentation).
>
>
> | Method | Optimized Traj. | Aug. | Motivation. |
> |--------|--------|--------|--------|
> | NoisyRollout  | Non-aug prompts  | Gaussian noise | Increase the CoT diversity(from aug. samples) to optimize the original prompts( non-aug.)   |
> | MiCo(Ours)  | Augmented prompts  | Crop, resize  | Get better CoTs from easier samples(weak aug.) to optimzed harder prompts(strong aug.) |
>
>
> **Concurrent work.**  Augmented-GRPO is only one of our contributions. Our primary contribution lies in exploring self-supervised signals to train rule-based reinforcement learning. Additionally, NoisyRollout was released in April, just one month before our submission deadline. Therefore, it can be considered concurrent work.
>
> ---
>
> ``W2: It is necessary to evaluate on more benchmarks to illustrate the superiority of the method and model, such as MuirBench, MMIU``
>
> **Results on more benchmarks.** Thank you for your suggestion. We would like to highlight that we have already included additional benchmarks (MuirBench, BLINK) involving multiple images in Table 3 of the main paper. Here, we further include results on MMIU.
>
> | Method | MuirBench | BLINK | MMIU |
> |--------|--------|--------|--------|
> | Qwen2.5VL-7B | 58.43 | 55.54 | 51.67 |
> | Ours |  60.53 |  57.23 | 54.02 |
>
> **Other benchmarks are less suitable.** In fact, MuirBench, BLINK, and MMIU are not ideal for evaluating multi-image reasoning. Although they involve multiple images, they often treat them independently within VLMs. For example, many questions are of the form “How many hands have gloves on?”, where the model does not need to link or compare across images. In contrast, VLM2-Bench is specifically designed to require linking fine-grained visual cues across images, which closely aligns with our objective.
>
> **Detailed sub-track analysis.** We have also provided detailed analysis of the sub-tracks within the general benchmarks in Table 4. The results show that our method brings significant improvements on tasks that truly require comparing different images, such as visual retrieval, semantic correspondence, and more.
>
> ---
>
> ``W3: Typo： LVAA-Thinking ``
> Thank you for your thorough review. We will make the necessary revisions and ensure all details are carefully verified.
>
> ---
>
> ``W4: In a sense, I think the results in Table 1 cannot explain the superiority of Mico. For example, Eureka is only trained in a single-image scenario, but it is only 3 points behind Mico, which is trained on multiple images.``
>
> **Cost for training data construction.** We would like to emphasize that our main insight lies in leveraging self-supervised signals to train RL models. We only utilize existing video and image editing datasets without any additional manual processing. In contrast, the main contribution of MM-Eureka is the construction of a training dataset that "features diverse knowledge domains with human-verified answers and solution processes" (as stated in their paper), which is significantly more costly.
>
> Even without any human annotation or verification, our method still achieves state-of-the-art performance with a margin of 3%. Imagine extending their data construction pipeline to cover fine-grained comparisons across multiple images—the cost would increase substantially (as labeling multi-image data is far more complex), and there is no guarantee that the resulting performance would surpass ours.
>
> **Eureka also include "multiple images".** The Eureka dataset also contains examples involving multiple sub-figures, such as comparing different diagrams in a long image or understanding three-view technical drawings. Although these are concatenated into a single image input, such compositions still help the VLM learn to link visual cues across different images.
>
> **We also get higer results on single-image task.** Reinforcement learning naturally possesses generalization capabilities. It is expected that training CoTs on single-image tasks can generalize to some extent to multi-image scenarios. However, by training directly on multiple images, our method also achieves higher performance on single-image benchmarks, such as HallusionBench, MMMU, etc., as shown in Table 3.
>
> **The performance is already high.** Compared with our baseline (Qwen2.5-VL-7B-CoT), we have already achieved a 12.93% improvement, even surpassing GPT-4o. When performance is already high, further improvements become quantitatively more difficult to obtain.
>
> **Concurrent works.**  Additionally, all these works are concurrent works. Eventhough,we add them to discussion and surpasses them with a clear margin(3%)

---

> > ### Author Response · Authors · 2025-08-04
> >
> > Dear Reviewer sVvm,
> >
> > We sincerely appreciate the time and effort you have spent reviewing our submission. We hope that our rebuttal has addressed your concerns adequately.
> >
> > If you have a chance, we would be very grateful for any further feedback you might have, or any clarification you may need from our side. We would be happy to engage in additional discussion if that would be helpful.
> >
> > Thank you again for your thoughtful review.

---

> > ### Author Response · Authors · 2025-08-05
> >
> > Dear Reviewer sVvm,
> >
> > Thank you once again for your valuable feedback on our paper. In response to your comments, we have included additional experiments and clarifications to better address your concerns.
> >
> > As the author-reviewer discussion period is drawing to a close, we would greatly appreciate it if you could let us know whether our responses have sufficiently addressed your questions.
> >
> > We are sincerely grateful for your time and thoughtful consideration.

---

> > ### Comment · Reviewer_sVvm · 2025-08-06
> >
> > Thank you for the author's response. After careful reading, I still hold the following views, and I believe this paper is too mundane to provide effective insights. Therefore, I maintain my original score, and I think a score of 3 is very appropriate for this article.
> >
> > Although the authors list the differences from NoisyRollout, they must acknowledge that they are still very similar. For example, crop, resize, and noise are all common techniques in image augmentation, and this minor difference is hardly convincing.
> >
> > The improvements on other multi-image benchmarks chosen by the authors are too marginal, with significant improvements only on the benchmark established by the authors themselves. This raises doubts about the effectiveness of the method.
> >
> > "We only utilize existing video and image editing datasets without any additional manual processing. In contrast, the main contribution of MM-Eureka is the construction of a training dataset that 'features diverse knowledge domains with human-verified answers and solution processes.'" This statement is not at all an explanation for why their method only shows marginal improvement over Eureka. If the authors believe that open-source datasets are of poor quality, they should collect high-quality data, such as a multi-image version of MMK12, which would make a greater contribution.

---

> ### Author Response · Authors · 2025-08-06
>
> Thank you for your valuable suggestions. We provide further clarifications below and would appreciate it if you could let us know whether your concerns are fully addressed. We are happy to engage in further discussion if needed.
>
> ``Although the authors list the differences from NoisyRollout, they must acknowledge that they are still very similar. For example, crop, resize, and noise are all common techniques in image augmentation, and this minor difference is hardly convincing``
>
> **The difference goes far beyond augmentation types.** In fact, one could argue that our use of data augmentation is fundamentally **opposite** to that in NoisyRollout. While they optimize their models on the non-augmented samples and use augmentation merely to improve rollout diversity, we directly optimize on augmented samples—because our goal and the function of augmentation are entirely different. They use augmentation as a tool for exploration; we use weak augmentations to generate easier questions that help guide learning.
>
> **NoisyRollout is a concurrent work.** Even though we maintain that our method is conceptually distinct (despite surface-level similarity), we kindly remind that NoisyRollout was developed concurrently and thus does not invalidate the novelty of our approach.
>
> ---
>
> ``The improvements on other multi-image benchmarks chosen by the authors are too marginal, with significant improvements only on the benchmark established by the authors themselves. This raises doubts about the effectiveness of the method.``
>
> **The improvement is not marginal.** Our method achieves +2.1%, +1.8%, and +2.4% improvements on MuirBench, BLINK, and MMMU, respectively—these are well-recognized, publicly available benchmarks for general image understanding. Note that such gains are substantial in the context of complex, diverse **image understanding** tasks that do not rely heavily on structured reasoning (unlike math datasets).
>
> **VLM2-Bench is not our own benchmark.** We kindly remind that VLM2-Bench is not "established by the authors themselves". We choose this benchmark because it is currently the only benchmark that focuses on linking fine-grained visual cues from multiple images.
>
> Moreover, as shown in the table below, our method outperforms other concurrent methods, including those trained on both single- and multi-image data, even though we train only on multi-image data:
>
> | Model               | MuirBench | BLINK  | Hallusion | MMStar | MMMU   |
> |---------------------|-----------|--------|-----------|--------|--------|
> | Qwen2.5VL-7B        | 58.43     | 55.54  | 69.50     | 64.06  | 54.11  |
> | MM-Eureka-7B        | 60.57     | 54.39  | 68.45     | 65.73  | 54.11  |
> | NoisyRollout-7B     | 59.61     | 56.07  | 66.66     | 65.66  | 54.55  |
> | VLAA-Thinker-7B     | 61.00     | 54.81  | 69.08     | 63.60  | 54.44  |
> | ThinkLite-VL-7B     | 57.62     | 55.81  | 72.97     | 66.80  | 53.55  |
> | **MiCo-7B**         | 60.53     | 57.23  | 69.61     | 65.60  | 54.77  |
>
> ---
>
> ``If the authors believe that open-source datasets are of poor quality, they should collect high-quality data, such as a multi-image version of MMK12, which would make a greater contribution.``
>
> We truly appreciate MMK12 and believe it is a valuable contribution to the community. We agree that data collection is one of the most effective ways to improve model performance.
>
> However, we also believe that building datasets is not the only way to advance research. Our paper explores a different direction: learning useful supervision without any human annotations, using self-supervised multi-image comparisons. We hope this work can inspire the community to consider alternative, scalable approaches for improving multi-image understanding.
>
> Furthermore, we would like to point out that MM-Eureka is also a concurrent work, and our approach is complementary to theirs. Our "contrastive comparison samples" can be directly incorporated into their data pipeline to further enhance multi-image understanding.

---

> > ### Author Response · Authors · 2025-08-07
> >
> > Dear Reviewer,
> >
> > Thank you again for your time and thoughtful feedback on our submission. We have provided detailed responses to your comments, and we truly appreciate the opportunity to address your concerns.
> >
> > As the discussion period is nearing its end, we would be very grateful if you could take a moment to revisit our responses and let us know whether they have resolved your questions, or if there are remaining points you would like us to clarify further.
> >
> > We are happy to engage in further discussion if needed. Thank you again for your time and consideration.

---

> > ### Comment · Reviewer_sVvm · 2025-08-08
> >
> > Thank you for the author's response. I acknowledge my misunderstanding regarding VLM2-Bench. However, I still have significant doubts about the effectiveness of the method. As the author mentioned, Mico is a model optimized for multi-image tasks, but on MuirBench and MMMU (the multi-image benchmarks mentioned by the author), it only shows very marginal improvements over MMEureka, which was trained only on single images (even with a training set consisting entirely of mathematics) - less than 1% improvement. This inevitably raises doubts about the method's effectiveness. Having substantial improvements on only one benchmark is insufficient to meet NeurIPS standards (for application-type papers).
> >
> > Secondly, I understand that the dataset is not the sole contribution. However, given the poor performance and the fact that the author's method fails to address generalization issues on other benchmarks, from the perspective of contributing a better model, the dataset is indeed what the author should focus on.
> >
> > Therefore, I believe this paper fails to meet NeurIPS standards in terms of both contributions and insights, and I maintain my score.

---

> ### Author Response · Authors · 2025-08-08
>
> Thank you for your response. We think the **lots of misunderstanding remains**, we hope the following clarification could address your concern.
>
> `` The effectiveness of the method over MM-Euraka.``
>
> **MuirBench, MMIU, BLINK are NOT real "multi-image".** Although they gives more than one images in the question, large portions of questions tackles different images independently like "How many synthetic images are there in totally?" "How many apples are there in these images". For these questions, models do not need to make comparisons among images.  Instead, they could understand these images independently one-by-one.
>
> **We bring significant improvements for real "multi-image" understanding.** We give more detailed sub-track analysis as follows, these REAL multi-image track are extracted from MuirBench, MMIU, BLINK. We demonstrate significant improvement in these tracks that requires linking visual cues among images.  However, as MM-Eureka is better at "fake multi-image" sub-tracks likes "counting", "diagram understanding", "math" ( these tasks don't require linking visual cues and are closer to their training set). In this way, we does not show significant advantages for the overall improvements.  As those  fake multi-image" sub-tracks are not actually evaluating multi-image understanding ability, we repeatedly emphasize the importance to use VLM2-Benchmark, which is more aligned with our motivations.
>
> | Model       | Semantic_Correspondence | Spatial_Relation | Visual_Similarity | Visual_Retrieval | Geographic_Understanding |
> |-------------|-------------------------|------------------|-------------------|------------------|--------------------------|
> | MM-Eureka    | 0.33             | 0.82     | 0.77     | 0.57       | 0.44                     |
> | MiCo      | 0.46  **(+13%)**            | 0.90 **(+8%)**     | 0.85 **(+8%)**      | 0.71 **(+14%)**       | 0.53 **(+9%)**                     |
>
> ``However, given the poor performance and the fact that the author's method fails to address generalization issues on other benchmarks``.
>
> **We give better results on single image tasks** as shown in the following table. We kindly ask **if you would challenge the contribution of MM-Eureka as it does worse than a method only trained on multiple images? and it fails to generalize to other tasks?**  We believe that this does not affect the great contribution of MM-Eureka as we focus on different domains. They give strong results on math (4% better than ours on MathVistas ) and we are good at fine-grained comparisons ( 4% better than  MM-Eureka on VLM2-Bench ). We both generalize to some single/multiple image benchmarks,  we give acceptable improvements compared with MM-Eureka for general image understanding.
>
> | Model         | Hallusion [11] | MMStar [6] | MMMU [39] |
> |---------------|---------------|------------|-----------|
> | MM-Eureka-7B  | 68.45         | 65.73      | 54.11     |
> | MiCo-7B       | 69.61         | 65.60      | 54.77     |
>
> **Our improvements are not marginal.**  We kindly remind that our method achieves +2.1%, +1.8%, +2.4%, +1.6% and improvements over-baseline on MuirBench, BLINK, and MMMU, MMStar.  And brings **6%** improvements on the benchmarks that mostly aligns with our motivation. We respectively remind  this could be claimed as "poor performance".
>
> **Please notice that our baseline is not MM-Eureka and it is a concurrent work.** We kindly remind that it is not proper to use MM-Eureka to reject our paper, as it is released on Mar 2025, which should be considered as a concurrent work. We really admire the great contribution of MM-Eureka, and we would also explore to combine MMK12 and our data for better results and acknowledge this great work in our open-sourced repos ( upon acceptance ) to increase its impact.

---

> > ### Comment · Reviewer_sVvm · 2025-08-08
> >
> > I believe the author has some misunderstandings. Using MM-Eureka is simply because it's a common baseline, and its performance on multi-image tasks listed by the author is nearly identical to Mico, which confuses me greatly.
> >
> > "MuirBench, MMIU, BLINK are NOT real 'multi-image.'" Does the author have any quantitative evidence to prove this point? If it's merely qualitative analysis, it's hard to convince me.
> >
> > "If you would challenge the contribution of MM-Eureka as it does worse than a method only trained on multiple images? and it fails to generalize to other tasks?" I think the author misunderstands, because multi-image training consumes more resources than single-image training. Single-image is a special case of multi-image, so it's not surprising that multi-image training can perform well on single-image tasks.
> >
> > The author may have misunderstood - by "marginal" I mean compared to other multimodal reasoning model baselines (e.g., MM-eureka on MuirBench, MMMU).

---

> ### Author Response · Authors · 2025-08-08
>
> We really appreciate your constructive discussion. We provide more informations as follows.
>
> ``MuirBench, MMIU, BLINK are NOT real 'multi-image.'" Does the author have any quantitative evidence to prove this point? If it's merely qualitative analysis, it's hard to convince me.``
>
> | Model         | MuirBench  | BLINK  | MMIU  |
> |---------------|---------------|------------|-----------|
> | Qwen2.5-VL-7B  | 58.43    | 55.54       |  51.67    |
> | MM-Eureka-7B  | 60.57     | 54.39      |   52.04   |
> | MiCo-7B       | 60.53         | 57.23     |  54.02    |
>
> **Statistics.** We start by giving the statistics for some "fake" multi-image tasks in MuirBench.  The following tasks do not require to make comparisons among images. They already count 52.3% of the totally question.
>
> | Task Name              | Ratio   | Example                                                                                                  |
> |------------------------|---------|----------------------------------------------------------------------------------------------------------|
> | Image text alignment   | 17.80%  | Which picture below better fits the description: ...                                                     |
> | Diagram Understanding  | 15.30%  | Which shape has an area of 6 square units? The shapes are made of unit squares.                  |
> | Counting               | 9%      | How many other garments besides a complete mitten pair are shown in each image?                         |
> | Scene Understanding    | 7.20%   | what colors are painted along the curb in the given images? |
> | Cartoon Understanding  | 3%      | What is the main content of this comic strip?                                |
>
> The statistics on BLINK is listed as follows, it is also not that suitable for multiple image understanding.
> | "Real multi-image"                 | Rows |
> |------------------------------------|------|
> | Multi-view_Reasoning               | 266  |
> | Semantic_Correspondence            | 279  |
> | Spatial_Relation                   | 286  |
> | Visual_Correspondence              | 344  |
> | Visual_Similarity                  | 271  |
> | Functional_Correspondence          | 260  |
> | IQ_Test                            | 300  |
> | Jigsaw                             | 300  |
>
> | "Single image Input"                | Rows |
> |------------------------------------|------|
> | Art_Style                          | 234  |
> | Counting                           | 240  |
> | Forensic_Detection                 | 264  |
> | Object_Localization                | 247  |
> | Relative_Depth (Single Input Image)                    | 248  |
> | Relative_Reflectance   (Single Input Image)              | 268  |
>
> MMIU contains many subtracks and the boundaries for "real/fake multi-image" is not that clear. It is not easy to give an accurate assessment. However, we could still find many tasks are  "fake multi-image" like "Forensic Detection","Visually Grounded Reasoning", "Text-to-Image Retrieval","Emotion Recognition""Multi-image Captioning", etc.
>
> The fact that previous benchmarks are not that suitable for evaluating the ability of linking visual cues is specifically pointed out and discussed in VLM2-Bench:  "existing benchmarks on multiple images and videos fall short in exploring this fundamental ability as they: (a) do not require explicitly linking visual cues across images or frames ." That is also the reason we find VLM2-Bench mostly aligns with our motivation.
>
> ``because multi-image training consumes more resources than single-image training. Single-image is a special case of multi-image, so it's not surprising that multi-image training can perform well on single-image tasks.
> ``
>
> We kindly remind that there are also many "multi-image" cases in MMK12, like they concatenate multiple diagrams together as a single image.  The key is teaching the model to compare and link visual cues rather than image number.
> The computation burden s similar with real  multi-image input, as Qwen preserves the original resolution of the images. Thus, the token numbers of similar. In this case, it is easy to understand that MM-Eureka could own generalization abilities  to some multi-image tasks.
>
> ``The author may have misunderstood - by "marginal" I mean compared to other multimodal reasoning model baselines (e.g., MM-eureka on MuirBench, MMMU).``
>
> Thank you for your clarification.  After explaining that MuirBench is not a real multi-image understanding benchmark. We would conclude that, compared with MM-Eureka,  we are 4% better on VLM2-Bench, 2.9% better on BLINK, and 2% better on MMIU, and generally better on "general single image understanding tasks". However, MM-Eureka is much better for math.
>
> ---
>
> Generally, we agree that it is important to convince the readers why previous benchmarks like MuirBench is not suitable and why we focus on VLM2-Bench.   We believe that this is the core reason for our misunderstand.
> Thank you for pointing out, and we would make it clearer in the revision.

---

> > ### Author Response · Authors · 2025-08-09
> >
> > Dear Reviewer,
> >
> > We have provided the requested statistics and sincerely hope you can take a moment to review our updated response.
> >
> > - We greatly appreciate your detailed feedback, which has helped us identify and clarify several misunderstandings. We will address all of these points thoroughly in the revision.
> >
> > - We would like to kindly reiterate that MM-Eureka is a concurrent work, and per the NeurIPS policy, we are not required to include it in our comparisons. Nevertheless, we fully acknowledge its valuable contributions and will follow your suggestion to provide a more detailed discussion of MM-Eureka, as well as explore ways to make our method compatible with this simple yet effective approach.
> >
> > - We believe that most major misunderstandings have already been addressed in our previous discussions. If you have any new concerns, we would be happy to provide further clarification.

---

### Decision · Program_Chairs · 2025-09-17

**Decision:**

Accept (poster)

**Comment:**

The paper received Accept, Borderline Accept, and two Borderline Reject ratings. The authors and the reviewers had an extensive discussion. The reviewers initially raised various concerns such as:
- Similarity to prior work such as NoisyRollout.
- Marginal improvement over prior work as shown in Table 1.
- Lack of generalization to other VLMs.
- Reliance on high-quality contrastive pairs.
- Comparison with Qwen2.5-VL-7B-CoT, which has a noticeable performance drop compared to the base version.

The authors submitted an extensive response, including additional analyses and new results. The works cited by the reviewers are primarily concurrent, so they do not provide grounds for rejection. The authors also provided comparisons with models specifically designed for reasoning and clarified that several existing benchmarks contain a significant proportion of single-image questions. After reviewing the paper, the reviews, and the rebuttal, the AC is convinced that the authors have addressed the reviewers’ concerns satisfactorily. Therefore, acceptance is recommended.